# scGraphformer: unveiling cellular heterogeneity and interactions in scRNA-seq data using a scalable graph transformer network
Xingyu Fan[1], Jiacheng Liu [1]✉, Yaodong Yang [1], Chunbin Gu[1], Yuqiang Han[1], Bian Wu[2], Yirong Jiang [3], Guangyong Chen [2]✉ & Pheng-Ann Heng[1]

The precise classification of cell types from single-cell RNA sequencing (scRNA-seq) data is pivotal for dissecting cellular heterogeneity in biological research. Traditional graph neural network (GNN) models are constrained by reliance on predefined graphs, limiting the exploration of complex cell-to-cell relationships. We introduce scGraphformer, a transformer-based GNN that transcends these limitations by learning an all-encompassing cell-cell relational network directly from scRNA-seq data. Through an iterative refinement process, scGraphformer constructs a dense graph structure that captures the full spectrum of cellular interactions. This comprehensive approach enables the identification of subtle and previously obscured cellular patterns and relationships. Evaluated on multiple datasets, scGraphformer demonstrates superior performance in cell type identification compared to existing methods and showcases its scalability with large-scale datasets. Our method not only provides enhanced cell type classification ability but also reveals the underlying cell interactions, offering deeper insights into functional cellular relationships. The scGraphformer thus holds the potential to significantly advance the field of single-cell analysis and contribute to a more nuanced understanding of cellular behavior.

scRNA-seq has emerged as a transformative tool in genomics, capable of comprehensive transcriptomic profiling at a cellular level since its first emergence[1]. scRNA-seq has been indispensable in deconvoluting the complexity of biological systems, delineating cell types, and tracking developmental trajectories[2]. The technique has led to breakthroughs across diverse scientific disciplines, from illuminating the heterogeneity of tumor microenvironments in cancer research[3] to dissecting the intricate cellular composition of the immune system[4] and unraveling the complexity of the brain[5]. Despite these advances, the analytical challenge posed by the high dimensionality and technical variability inherent in scRNA-seq data persists, hindering the extraction of meaningful biological insights.

In response to these challenges, the adoption of deep-learning methodologies, such as scVI[6], scBalance[7], and ACTINN[8], has enhanced the understanding of cellular heterogeneity. Moreover, machine learning algorithms, such as scmap[9], Phenograph[10], MAGIC[11], and Seurat[12], lean on k-nearest neighbor (kNN) algorithms to infer cell-cell relationships,

delineating latent cellular networks. In contrast to other machine learning techniques, kNN's distinctive attribute allows for the comprehension of cellular connections within heterogeneous groups through network construction. Despite their utility, the rapid evolution of sequencing technologies and the resulting data expansion have exposed the limitations of kNN-based methods, particularly in their ability to scale and resolve intricate cellular relationships in large and complex datasets.

GNN have emerged at the forefront of computational biology as a promising solution to these limitations[13-17]. Through integrating gene expression with graph learning, a cellular network maps out individual cells as nodes, with edges symbolizing relationships between these cells. Despite robust intercellular correlations, graph structures within inter-cells of scRNA-seq data are conspicuously absent, provoking a reliance on kNN to synthetically construct a predefined noisy cellular network[18]. However, applying a predefined graph may restrict graph learning performance within scRNA-seq, ultimately limiting its potential.

[1]Department of Computer Science and Engineering, The Chinese University of Hong Kong, Hong Kong, China. [2]Zhejiang Lab, Hangzhou, China. [3]Department of Chemistry, Zhejiang University, Hangzhou, China. ✉e-mail: jiachengliu@cuhk.edu.hk; gychen@zhejianglab.com

To overcome the limitations inherent in current methodologies, we propose scGraphformer, a cutting-edge approach that integrates the transformative capabilities of the Transformer model with the relational inductive biases of GNN. This graph transformer network abandons the dependence on predefined graphs and instead derives a cellular interaction network directly from scRNA-seq data. By treating cells as nodes within a graph and iteratively refining connections, scGraphformer captures the full spectrum of cellular relationships, allowing for a more nuanced understanding of cell type. scGraphformer leverages the Transformer's self-attention mechanism to process information across the entire cellular network, revealing intricate patterns and dependencies that were previously obscured. This allows for the identification of key genes and their dynamic interactions, facilitating a comprehensive exploration of cellular heterogeneity. Our model is distinguished by its ability to establish biologically meaningful cell-cell relationships without relying on a hypothesis-driven predefined structure. Instead, it learns these relationships from the data itself, thereby avoiding the introduction of noise and bias associated with kNN graphs. Furthermore, scGraphformer is designed to be inherently scalable, tackling the challenges posed by large-scale scRNA-seq datasets. The model's ability to process and integrate vast amounts of gene-level information across cells positions it as a powerful tool for unlocking the latent structure of cellular networks. Our approach offers a novel solution for single-cell transcriptomic analysis, offering the potential to elucidate previously unrecognized cellular behaviors.

In this study, we validate the effectiveness of scGraphformer through extensive evaluations of diverse scRNA-seq datasets. Our results demonstrate that scGraphformer not only excels in cell type identification but also in revealing the underlying cell interactions, providing deeper insights into the functional relationships that govern cellular behavior. By pushing the boundaries of single-cell analysis, scGraphformer paves the way for a more profound understanding of the complex tapestry of life at the cellular level.

## Results

### The architecture of scGraphformer

The scGraphformer architecture employs transformer-based graph neural networks to provide accurate and scalable annotations of cell types. It stands out in its ability to dynamically construct an inter-cell relationship network through a refinement process that enhances the biological topological structure inherent in the cell graph. This innovative approach leads to the discovery of latent cellular connections, which are then harnessed to achieve precise cell type annotations. The scGraphformer framework is composed of two key components: a specially designed Transformer module and a cell network learning module. The Transformer module is adept at discerning latent interactions among genes, which in turn influence cellular connectivity. The cell network learning module is responsible for constructing a nuanced cell relationship network (Fig. 1). Unlike conventional methods that typically depend on predefined graph structures, scGraphformer is distinctive in its ability to learn the cell graph's structure directly from the raw scRNA-seq data, allowing for the continuous refinement of cell-to-cell connections and leading to more accurate cell type annotations.

The scGraphformer initially processes scRNA-seq data through standard preprocessing steps, which include the removal of low-quality cells and genes, normalization, and the selection of highly variable genes (HVGs). However, unlike other tools that adhere to a fixed count of variable genes, scGraphformer uniquely tailors the selection based on the dimensionality of the expression matrix, with the expectation that this approach will preserve a greater amount of genetic information, even when dealing with high-dimensional expression profiles. The data is then transformed into a graph structure without predefined edges, where each node symbolizes a cell, and node features represent the selected HVGs. It is also possible to augment scGraphformer with existing high-quality cell graphs. In this study, we opt for a kNN graph, a common choice in single-cell analyses. The connectivity in this graph is quantified by the distance between pairs of cells (Fig. 1a). It is important to note that the incorporation of such a graph into scGraphformer is not mandatory; our findings demonstrate that the integration of existing, potentially noisy, cell graphs does not invariably lead to enhanced performance.

Upon constructing the initial graph, scGraphformer encodes the cell node features into distinctive gene representations through a multilayer perceptron (MLP). This step is crucial for capturing the diverse expression patterns across different cell populations, particularly aiding in distinguishing rare cell types. Subsequently, these gene representations are fed into the scGraphformer layer, which includes a specifically engineered Transformer module coupled with a cell network learning module. The Transformer, originally a paradigm-shifting architecture in the realm of natural language processing due to its proficiency in detecting distant dependencies within text, has been meticulously reconfigured to address the intricacies of scRNA-seq data. The Transformer layer within scGraphformer employs a multi-head attention mechanism to discern the influence of gene-gene interactions on cellular relationships and to compute gene attention scores contextualized by cellular phenotypes. This mechanism adeptly identifies genes that play lesser roles in cell development. Further enhancing the architecture, the scGraphformer layer includes a cell network learning structure that dynamically updates the cell relationship network's topological structure. This module amalgamates an aggregation mechanism with a computation mechanism, integrating the learned gene-gene interactions and the evolving cell network to continually refine the graph. Through this process, scGraphformer not only accurately annotates cell types but also offers an unprecedented view into the complex network of cellular interplay (Fig. 1b and Supplementary Fig. A1).

The Transformer module within scGraphformer is re-engineered to suit the nuances of biological data, with its Query, Key, and Value sub-modules receiving a bioinformatics-centric redesign. To enhance the integration of the attention mechanism into scRNA-seq data, these three parameters have been given more biological meaning. In scGraphformer, the Query module utilizes global information in gene representation to provide insights into the impact of gene interactions on cell phenotype. The Key module captures dependencies across multiple cells and helps avoid artificial biases in gene-gene interactions. The Value module provides contextualized representations of each cell. And the Key and Value modules also contribute to the construction of the cell graph. Regarding the computation process of Query, Key, and Value, the challenge lies in capturing gene expression patterns related to cell-specific regulatory signals. The latent biological connection between cells is primarily driven by the expression of relevant genes. Therefore, the computational process of Query, Key, and Value is designed to determine the final attention weights of gene representations (Fig. 1c). When a predefined graph structure is added (which is optional in our framework), a combination of Query and Key will be added as GCN (Graph Convolutional Network)-based propagation to the all-pair cell network. These attention weights are leveraged to perform cell-type annotation with proven precision. The scGraphformer layer utilizes a multi-head attention mechanism, consisting of multiple self-attention layers, to understand how gene-gene interactions influence cellular neighboring relationships and compute the attention scores of genes in the context of cellular phenotype. This attention mechanism effectively filters out genes that are less important in cell development.

### scGraphformer provides better performance than other single-cell classification methods in intra-dataset

To evaluate the performance of scGraphformer, a series of intra-dataset evaluations were conducted across 20 diverse datasets (Supplementary Table B1). The classification capabilities of scGraphformer were benchmarked against seven state-of-the-art computational strategies commonly utilized in scRNA-seq cell annotation: CellTypist[19], scVI[6], scmap-cluster[9], scmap-cell[9], ACTINN[8], scBert[20], TOSICA[21], scType[22] and scBalance[7].

As illustrated in Fig. 2a, scGraphformer demonstrated a high proficiency in accurately identifying cell types, showcasing its robustness across the varied datasets (Supplementary Table C5). Figure 2b further illustrates that scGraphformer surpassed other methods in accuracy for most of the

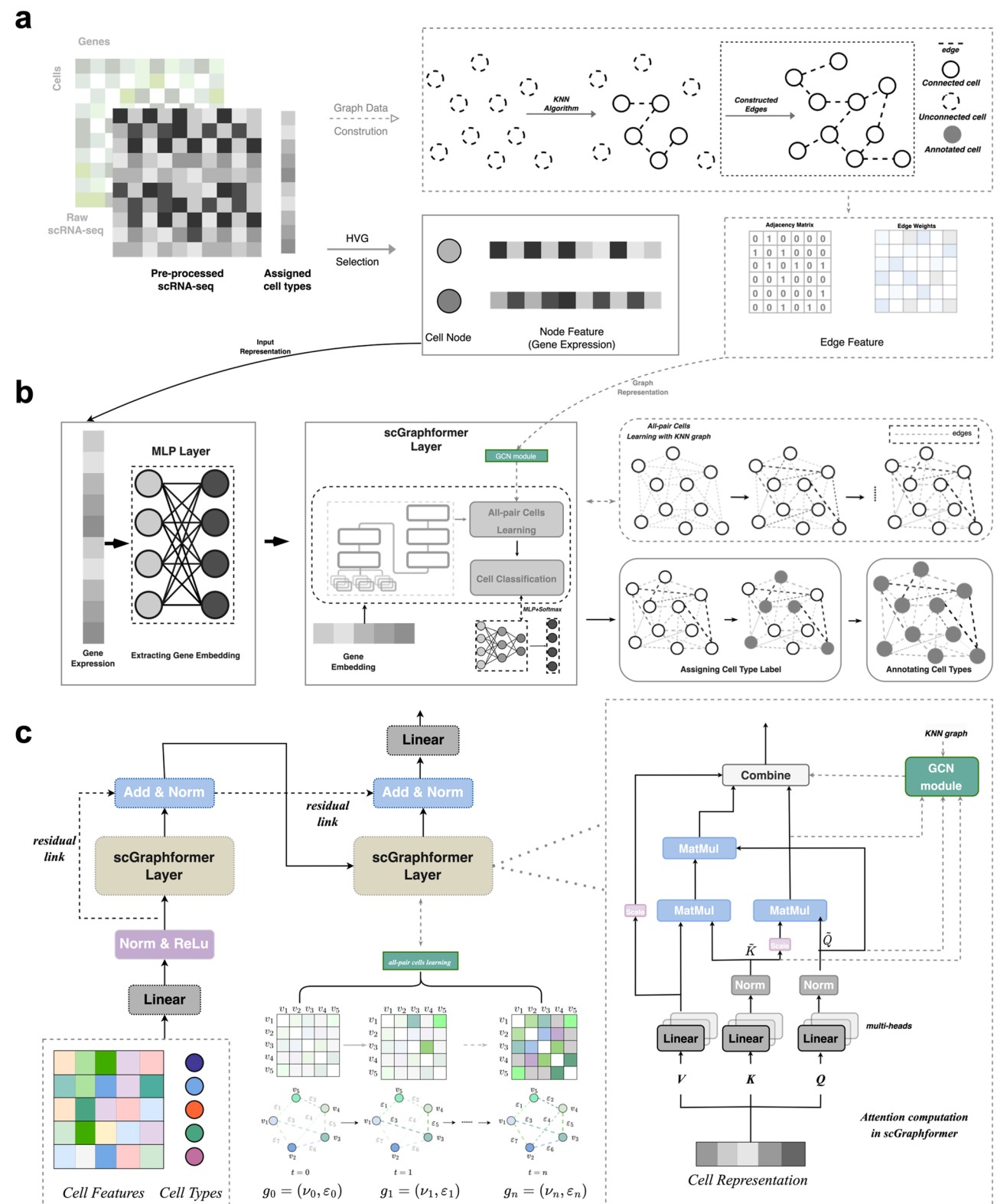

**Fig. 1 | Workflow of the scGraphformer approach for single-cell transcriptomic data analysis. a** Data preparation: Starting from raw single-cell RNA sequencing (scRNA-seq) data, scGraphformer employs standard normalization to preprocess the expression matrix. Cell nodes are defined by selecting highly variable gene (HVG) features. The pre-processed cell nodes with their HVG features will be input to the model. The pre-constructed graph structure $\mathcal{G}$ by kNN algorithm serves as a relational bias and is not compulsory. **b** scGraphformer pipeline: The normalized gene expression data undergo an initial transformation through a multilayer perceptron

(MLP) layer to obtain gene embeddings. These embeddings are then processed by scGraphformer layers, which learn the all-pair connected network and update the cell graph topology. While kNN graph is used, the model will assign adjacent nodes within the $\mathcal{G}$ with proper weights in the all-pair network. **c** scGraphformer architecture: Detailed schematic of the scGraphformer framework illustrating the sequence of operations within the scGraphformer layers. The left shows the model's architecture. The right shows the attention computation in scGraphformer layer, and the dashed lines show the computation process when adding a predefined graph structure.

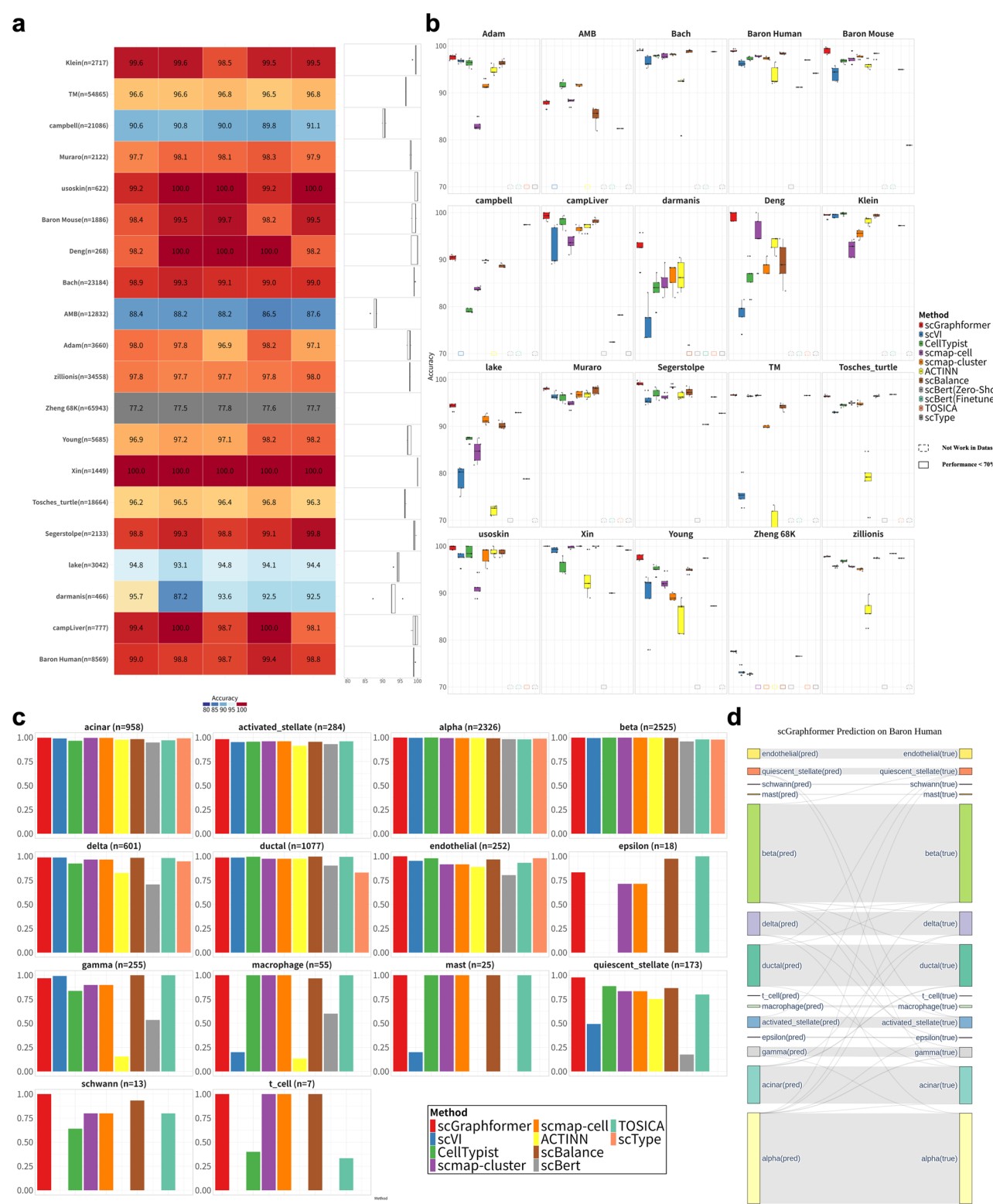

datasets, reaffirming its effectiveness in the realm of scRNA-seq cell annotation (Supplementary Table C2). Concurrently, we intentionally selected several complex datasets such as campbell[23], zillionis[24], and Zheng 68K[25] to test scGraphformer's performance in these challenging scRNA datasets, and the results show it performs well, which gives us confidence for further training on large-scale atlas. Among all these datasets, the performance of scGraphformer also shows its good stability across fivefold experiments where the results consistently showed minimal variation. Some methods also show outstanding performance on several datasets, but some of them are not sensitive to minor cell types and struggle to identify rare populations. Meanwhile, several methods faced limitations in our evaluation. For example, scBERT's performance was hindered by the absence of specific genes in some datasets. TOSICA's applicability was restricted due to its support for only human and mouse GOBP. scType's annotations sometimes mismatched reference cell types, requiring manual adjustments. These constraints highlight the challenges in developing universally

**Fig. 2 | scGraphformer shows its impressive performance in annotating cell types in intra-dataset. a** Heatmap portraying the accuracy of five replicates for each scRNA-seq dataset, where color depth corresponds to accuracy level. The adjacent boxplot illustrates the distribution of accuracies per dataset, delineating the median (central line), interquartile range (box), and any outliers (dots). The n number represents the number of cells in each dataset, and the number within each box represents the accuracy of annotation in each dataset. **b** A comparative assessment of scGraphformer and alternative methods within intra-datasets, each subjected to quintuplicate evaluations to confirm consistency. Boxplots illustrate scGraphformer's comparative annotation performance, revealing its consistently higher median accuracy and reduced variability, particularly in complex datasets like Zheng 68K,

TM, and zillionis. The solid frames indicate that the method's performance on the respective dataset is below 70%, while dashed frames signify that the method is not applicable to that particular dataset. Each experiment is conducted five times, with each point representing an individual result. The error bars illustrate the variability across these five independent trials (details of each dataset are shown in Supplementary Table B1). **c** A cell-type-specific precision analysis within the Baron Human dataset, demonstrating scGraphformer's heightened accuracy in identifying scarce cell types relative to competing approaches. **d** Sankey diagram representing scGraphformer's annotation accuracy within the Baron Human dataset, showcasing the correspondence between predicted and true cell type classifications.

applicable cell-type annotation tools. And we use dashed box to represent their absence in that dataset.

To rigorously assess scGraphformer's proficiency in discerning minor cell populations and to establish its comparative advantage, we scrutinized the precision of cell type identification. Illustrated in Fig. 2c and Supplementary Table C6, our analysis specifically focuses on the classification accuracies within an imbalanced dataset, Baron Human[26]. While most models exhibited a high accuracy in classifying major cell types like alpha and beta cells, the identification of rare types such as mast cells, epsilon cells, and Schwann cells posed a significant challenge. However, scGraphformer notably excelled in this area, reliably classifying these less abundant cell types-a feat that proved challenging for several competing models, some of which failed to detect them entirely. This finding underscores scGraphformer's enhanced capability to handle diverse cell populations and its adeptness in classifying minor cell types in imbalanced datasets. Additionally, the Sankey diagram in Fig. 2d distinctly presents scGraphformer's classification outcomes for each cell type in the Baron Human dataset.

In conclusion, scGraphformer demonstrates exceptional performance in baseline annotation tasks, characterized by its stable proficiency in accurately identifying both major and minor cell types. This versatility and reliability mark scGraphformer as a valuable tool in the realm of scRNA-seq analysis.

## scGraphformer is robust in inter-dataset evaluation

The robustness of cell annotation methods in scRNA-seq, particularly across varied datasets (inter-dataset evaluation), is crucial. Practitioners often train models using reference data and apply them to query data derived from disparate platforms or protocols. A major hurdle here is the batch effect, a systematic non-biological variation caused by differences in sequencing methods that can significantly undermine annotation precision. Our study focuses on the resilience of several annotation methods in the face of batch effects, aiming to ensure consistent cell type identification across datasets. We utilized seven datasets from the PbmcBench collection[27], sequenced with diverse protocols, to conduct an exhaustive cross-platform evaluation. We scrutinized the performance of seven annotation methods, with particular emphasis on their comparative efficacy to scGraphformer in identical experimental setups.

Preliminary results, illustrated in Fig. 3a, b, demonstrate scGraphformer's proficiency in maintaining annotation accuracy across datasets when trained on 10Xv2 and tested on other platforms, and when trained on 10Xv2, scGraphformer achieved a mean accuracy of 95.46%, which is approximately 2% higher than the average of second-best methods (CellTypist: 93.66%, details shown in Supplementary Table D5). During training, genes that were not expressed in both training and testing data were excluded from the feature space, thereby enhancing the method's resilience to batch-induced variability. The results indicate that scGraphformer's ability to overcome batch effect. Furthermore, to compare its performance with other methods under batch effect, we evaluate the annotation performance of all pairwise train-test combinations between 7 protocols on the left methods. All 49 experiment results are summarized in Fig. 3c, and the Adjusted Rand Index (ARI) score is used as the evaluation metric. We set a training proportion of 80% in reference datasets for realistic situations (details in Supplementary Appendix D and Table D8). Besides ARI score, we

also used accuracy score, Normalized Mutual Information (NMI), F1-score to compute the annotation performance (Supplementary Figs. D2, D3 and Table D7). Under each metric, scGraphformer achieves better overall performance, which has better median and mean values across all experiments. (Supplementary Table D10 and Fig. 3d). This variance underscores the method's adaptability to diverse dataset characteristics and sequencing platforms. The results show that scGraphformer achieved a good performance across all experiments, and there is not much difference from other state-of-the-art methods. Moreover, scGraphformer's performance at rare cell type identification was also demonstrated through cross-platform experiments, where we employed Uniform Manifold Approximation and Projection (UMAP) for visualization purposes while training on Smart-seq and testing on 10Xv3 (Fig. 3e and Supplementary Table D9), and it visualizes the clustering result of scGraphformer with the original true cell types and predicted cell types. This aspect of the analysis highlighted scGraphformer's superior performance in accurately annotating minor cell populations, a critical capability given the frequent occurrence of imbalanced datasets in scRNA-seq studies.

Meanwhile, we explored the performance of training on fused datasets to simulate real-world scenarios where biologists often use mixed data for annotating newly sequenced datasets. For each evaluation of a single sequencing dataset, we integrated the remaining six datasets into a unified fusion dataset. This approach aligns with practical applications where datasets are not strictly separated by sequencing technology. Figure 3b illustrates the comparison between training on fusion data versus single data across different sequencing platforms. The results consistently demonstrate that training on fusion data outperforms single data training across all platforms. These findings validate our hypothesis that mixed data can enhance annotation performance and contribute to more accurate cellular identification. Detailed results and statistical analyses are provided in Supplementary Appendix D, Tables D11 and D12.

Taken together, scGraphformer demonstrates robustness and adaptability in inter-dataset cell type annotation. Its ability to accurately pinpoint minor cell populations and counteract batch effects makes it a valuable tool for scRNA-seq analysis.

## scGraphformer can accurately model the subtle cellular diversity in the mouse brain

To demonstrate that scGraphformer can effectively discover cell subtypes within major cell types that appear to be identical. We use two mouse brain scRNA-seq datasets, namely Zeisel[28] and Rosenberg[29], each comprising over 100,000 cells (Zeisel: 145,954 cells, Rosenberg: 133,435 cells). In this study, we utilized Zeisel as a reference dataset and then used the trained model to annotate the Rosenberg dataset. Shown in Fig. 4b, scGraphformer achieved the highest mean accuracy of 95.210% with a standard deviation of 0.710 which indicates not only superior performance in terms of accuracy but also a reasonable consistency across different runs. The second-best model, scmap-cell, achieved 94.40%, approximately 1% lower than scGraphformer. This performance gap underscores scGraphformer's effectiveness in cell-type annotation tasks (Details in Supplementary Table E13). In Fig. 4c, the result shows scGraphformer's robust performance in large-scale and heterogeneous datasets, particularly its ability to accurately classify a wide range of cell types, from the most to the least prevalent. The model's proficiency in

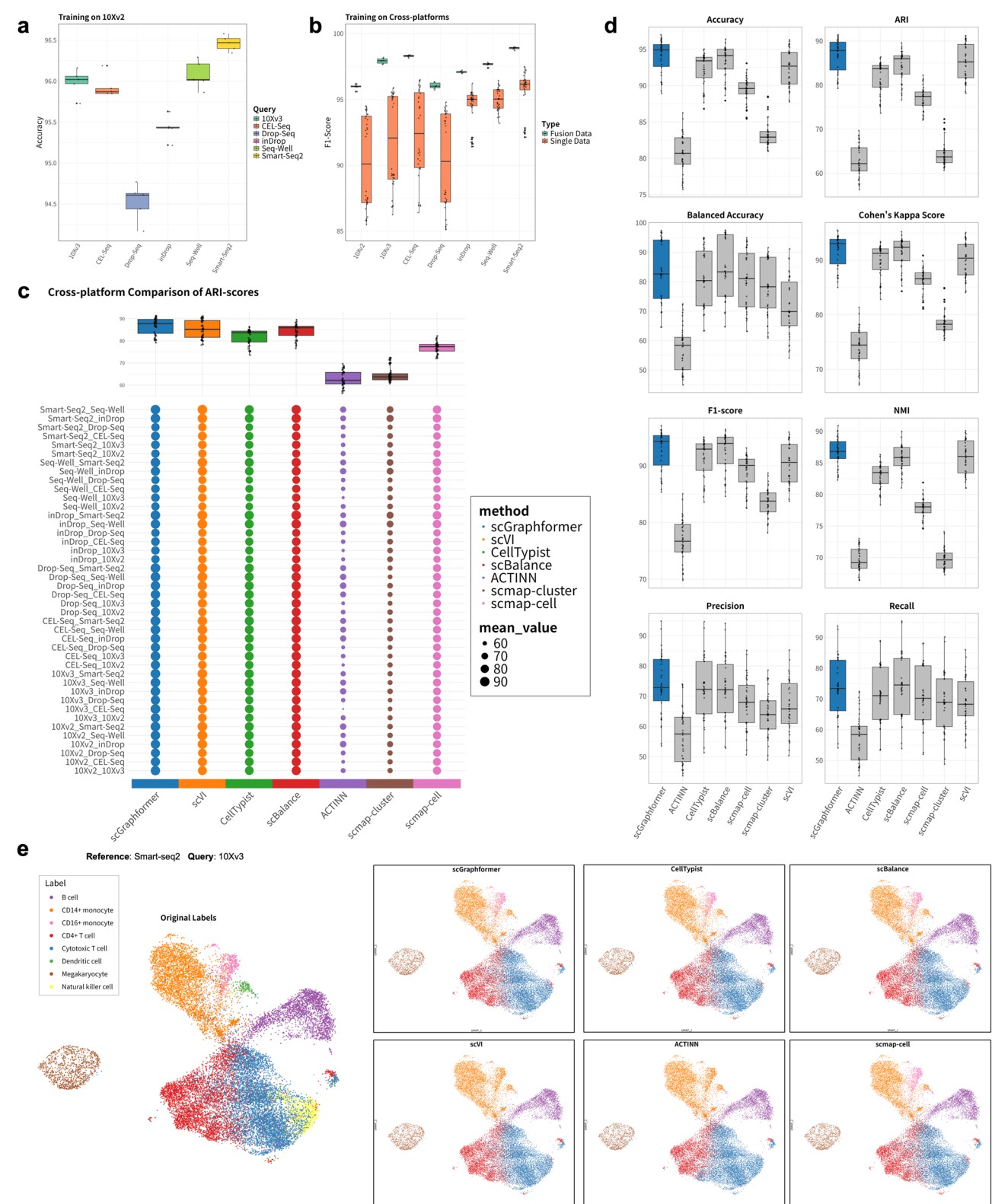

handling major cell types like neurons, as well as its ability to identify less represented groups such as macrophage cells which have a lower proportion within the dataset, highlights its potential on the imbalance dataset again. However, it shows its poor identification ability on brain pericyte cells and endothelial cells, where we found the model confused two cell types as indicated by the extracted confusion matrix. This difficulty can be attributed to the high phenotypic and functional similarity between these two cell

types, which often results in overlapping gene expression profiles. Pericytes and endothelial cells are integral components of the blood-brain barrier and share several molecular markers, complicating their discrimination in single-cell RNA sequencing data. Then, to better compare scGraphformer's performance on such a large and imbalanced cross-platform dataset, we also tested the performance of several methods on each cell type. As shown in Fig. 4a, we evaluated their identification results and showed their confusion

**Fig. 3 | Comparison of scGraphformer performance across different scRNA-seq protocols. a** Performance of scGraphformer when trained on 10Xv2 ($n = 23{,}154$) protocols, and applied to annotate cell types from other protocols (10Xv3 ($n = 19{,}690$), CEL-Seq ($n = 19{,}754$), Drop-Seq ($n = 23{,}154$), inDrop ($n = 21{,}832$), Seq-Well ($n = 18{,}966$), Smart-Seq2 ($n = 18{,}886$)). **b** Comparative performance of scGraphformer trained on single-platform data versus fusion data across various cross-platform scenarios. The *x*-axis represents different query datasets, while the *y*-axis shows the F1-Score, ranging from 85 to 100. Boxplots depict the statistical distribution of results for each dataset and data type. Green boxes represent models trained on fusion data, which integrates data from all sequencing technologies except the one being queried ($n$ is the total number of cells from those included sequencing technologies). Orange boxes represent models trained on single-platform data. **c** Overall annotation ARI scores of scGraphformer in comparison to other methods, with each train-test experiment pair labeled as "Train Dataset_Test Dataset." Accuracy scores serve as the primary metric of evaluation, with 42 distinct train-test combinations illustrated. The accompanying boxplot provides a summary of the comparative performance. The results show that scGraphformer performs better in the overall ARI score. **d** Comprehensive comparison of performance metrics across all cross-platform experiments ($n = 42$ experiments) for different methods. The error bar represents the variability across 42 experiments under each evaluation metric. Eight metrics are displayed: Accuracy, ARI, Balanced Accuracy, Cohen's Kappa Score, F1-score, NMI, Precision, and Recall. Each subplot uses boxplots to show how the seven methods (scGraphformer in blue, others in gray) perform across all experiments. scGraphformer consistently outperforms other methods in most metrics, especially in Accuracy, ARI, F1-score, and NMI, showing both higher median values and less variation. **e** UMAP visualization using scanpy displaying the classification of cell populations within the 10Xv3 dataset, annotated according to predictions from various methods, all of which were initially trained using the Smart-Seq PBMC dataset. scGraphformer shows a particular strength in the precise identification of rare cell types, as indicated by the distinct and accurate annotation of clusters.

matrix. Comparisons show that scGraphformer is better for processing large datasets in cross-platform experiments even if the proportion of cell types varies too much. It's particularly good at identifying major brain cell types like astrocyte cells, neurons, and oligodendrocyte cells. Compared to other methods, scGraphformer makes fewer mistakes in mixing up similar cell types. One common challenge for all methods is telling the difference between endothelial cells and brain pericytes. Even scGraphformer sometimes mixes these up. In comparison to scmap-cluster and scmap-cell, scGraphformer appears to have reduced off-diagonal elements, suggesting fewer misclassifications. However, scmap-cell shows comparable performance for certain cell types, particularly neurons and microglia. scGraphformer still has room for improvement in identifying certain cell types.

## scGraphformer can be applied to large-scale datasets to identify multiple cell types

As sequencing technologies rapidly advance and find commercial applications, the sheer volume of scRNA-seq data is burgeoning, necessitating scalable annotation tools. In response, we investigate the performance of scGraphformer in annotating large-scale atlases. We first evaluated our model using the Zheng 68K dataset, a well-established benchmark for large-scale studies. Subsequent assessments were conducted on two comprehensive datasets: the Covid-19 Immune Atlas, comprising 1,462,702 cells (shown in Fig. 5e), and the Human Neocortex Atlas, which includes 638,941 cells.

Given the extensive scale of expression profiles and the fundamental requirements of graph learning, our study extends to evaluating scGraphformer's efficacy on large heterogeneous graphs. To benchmark scGraphformer's performance, we compared it with scBalance, and scVI, which epitomize the spectrum of deep learning, traditional machine learning, and canonical cell annotation approaches in handling large-scale datasets. To mitigate the influence of less informative genes, we employed a HVG selection strategy. We experimented with different selection thresholds (3000, 4000, and 5000 genes) to identify the configuration that yields optimal annotation outcomes. Also, our evaluation needs a lot of computer memory to work. To make it run efficiently, we tested different numbers of genes: 3000, 4000, and 5000. We wanted to find the best number that gives good results without using too much memory. After testing, we found that using 4000 genes works best. Furthermore, we assessed scGraphformer's versatility by training it on the Covid-19 atlas and subsequently testing it on distinct cell type settings.

The cell type distribution in Zheng 68K, as shown in Fig. 5c, is notably imbalanced. The UMAP visualization in Fig. 5b illustrates the difficulty in distinguishing original cell types by quantity. Figures 2b and 5d reveal that scGraphformer excels in accuracy over other methods and performs notably well with minor cell types. In Zheng 68K dataset, as evidenced by the Fig. 5d, scGraphformer demonstrates stability and strong performance across most cell types.

In a comparative analysis between the Covid Atlas and the Human Neocortex Atlas, scGraphformer exhibits exceptional performance. For the Covid Atlas (Fig. 5f and Supplementary Fig. E2), as depicted in Fig. 5e, scGraphformer achieves a mean accuracy of 76.05%, outperforming Cell-Typist, which scored 72.5%. This highlights scGraphformer's adeptness in navigating the intricate cellular composition of the Covid Atlas. The Human Neocortex Atlas, however, saw all three methods achieving high accuracy, with scGraphformer slightly leading at 93.24%, compared to the close results from scBalance. The narrow accuracy margins in this atlas, which is characterized by distinct cellular composition and expression patterns, suggest a less challenging environment for cell type annotation and, consequently, a subtler display of scGraphformer's strengths. Complementing these accuracy metrics, Fig. 5g offers valuable insights into the computational efficiency of scGraphformer compared to other leading methods. The runtime analysis across three datasets of increasing complexity—Zheng68k, Covid Atlas, and Human Neocortex Atlas—reveals scGraphformer's superior performance in terms of processing speed. As we progress to the more complex Covid Atlas and Human Neocortex Atlas, the efficiency gap widens significantly. scGraphformer maintains consistently low runtimes, even as dataset complexity increases, whereas both CellTypist and scBert show substantial increases in processing time.

Despite this, the Human Neocortex Atlas predominantly comprises rare cell types, underscoring the importance of effective annotation tools. The UMAP visualization of scGraphformer's annotation results, shown in Fig. 5d, further emphasizes its proficiency in identifying rare cell types within large-scale scRNA-seq datasets.

This study presents an exploration of scGraphformer's annotation abilities within large-scale scRNA-seq datasets, underscoring its potential as a practical and efficient tool for managing substantial genomic data volumes.

## Elucidating cellular dynamics in the campLiver dataset via scGraphformer

We applied the scGraphformer to the campLiver[30] dataset and uncovered complex cellular interactions. The attention mechanism facilitated the identification of cells with potential interactions or similarities, as delineated through the attention maps. Through t-SNE (t-distributed Stochastic Neighbor Embedding) dimensionality reduction, we observed distinct cell clusters (Fig. 6a), indicating diverse cell populations. The bar chart (Fig. 6b) quantified the distribution of these cell types. Meanwhile, we plot a scatter plot (Fig. 6c) that represents the cell type in each index to help find clues in the attention matrix.

The scGraphformer is characterized by a two-layered architecture, resulting in two distinct attention matrix layers. The heatmaps corresponding to the attention matrices of both layers, as shown in Fig. 6d, reveal focal points that hint at possible cellular interactions. In contrast, the kNN connectivity matrix, which is generated based on expression values alone, serves to discern these interactions. Our analysis has led to several notable findings. For instance, the attention matrix from the first layer reveals a pronounced attention value between definitive endoderm and immature hepatoblast cells (red box in the attention matrix), which the kNN connectivity matrix does not capture. This observation aligns well with the

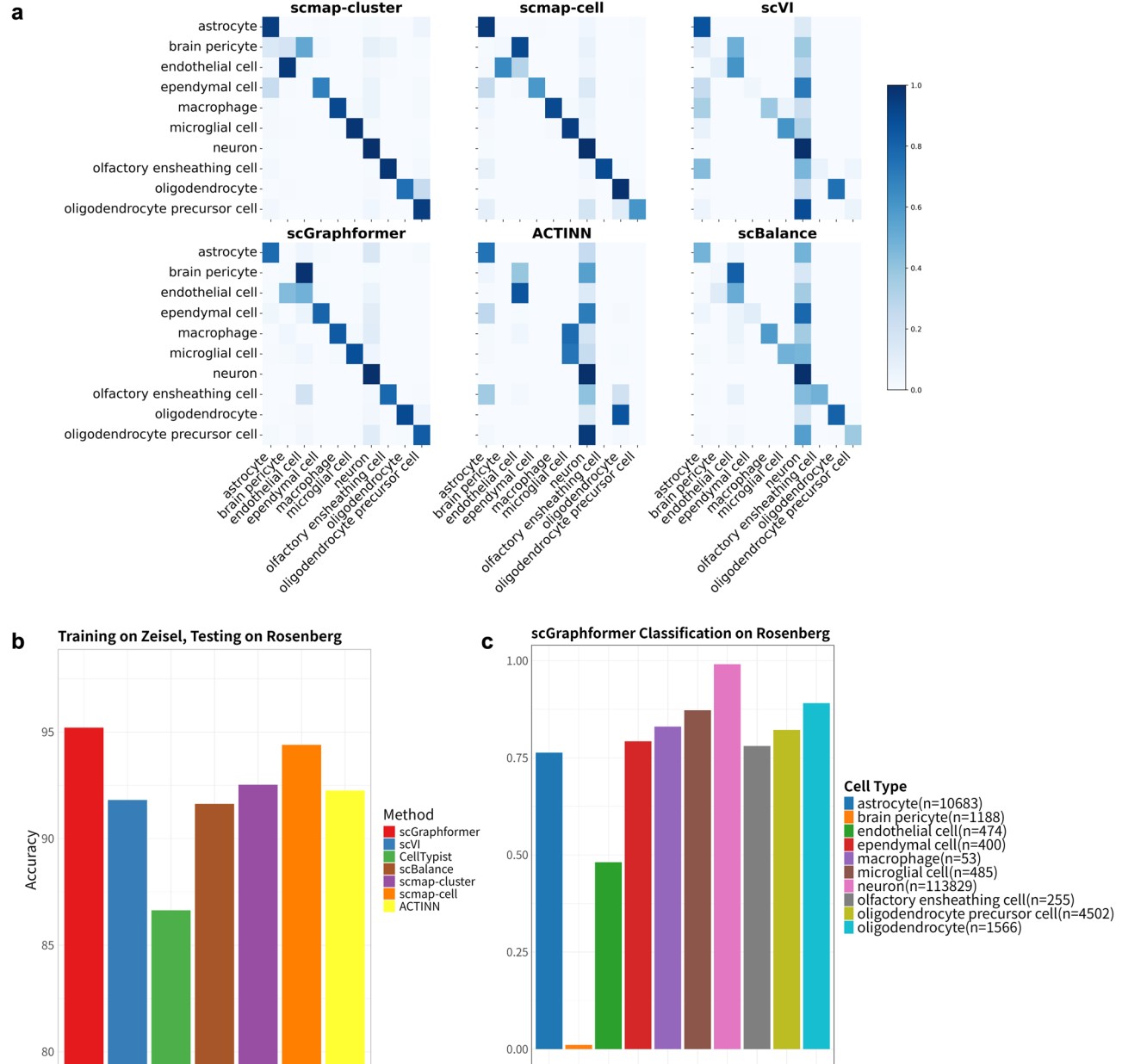

**Fig. 4 | Performance of scGraphformer on large-scale inter-dataset analysis.**
**a** Heatmap represents the confusion matrix of classification results for Rosenberg dataset cell types, with a side-by-side comparison to other methods. The intensity of the colors correlates with the quantity of cells accurately or erroneously classified. scGraphformer shows a clear advantage in correctly identifying cell types, especially within smaller subsets, demonstrating its precision and the breadth of its efficacy compared to other methods. To keep visual clarity and organization, CellTypist is excluded since it performs the worst. **b** Bar plot with error bars demonstrating the superior performance of scGraphformer when trained on the Zeisel ($n = 145,954$)

dataset and tested on the Rosenberg ($n = 133,435$) dataset, in comparison to other methods over five repeated runs. Each bar signifies the average accuracy for a given method, while error bars denote the standard deviation, illustrating the consistency and reliability of scGraphformer, as indicated by its higher mean accuracy and more concentrated error bars. **c** Bar plot illustrating scGraphformer's classification accuracy for each cell type within the Rosenberg dataset, alongside a legend indicating the count of cells per type. scGraphformer maintains high accuracy levels across various cell types, including those that are less represented.

direct developmental lineage shared by these two cell types, thus validating their established biological link[31]. The attention map further indicates a strong connection among hepatic endoderm, immature hepatoblasts, and mature hepatocytes (yellow box in the attention matrix), which collectively play a critical role in the hepatic lineage and its functions[32]. Additionally, the progression from immature hepatoblasts to mature hepatocytes is evidenced by the corresponding attention scores (orange box in the attention matrix), underscoring the developmental trajectory between these stages[31]. Collectively, these insights underscore the scGraphformer's proficiency in detecting potential cellular interactions.

To corroborate the findings suggested by the attention matrixes, we conducted a thorough analysis of gene expression profiles, which are depicted in Fig. 6e. This analysis unveiled gene expression patterns that align with the proposed cellular interactions. For instance, the gene KRT19, known for its expression in epithelial cells and utilized as a marker for immature hepatoblasts[33], also showed a significant expression in definitive endoderm cells, consistent with the attention matrix results. Furthermore, although FGB, the fibrinogen beta chain, is not a conventional marker for specific cell types, its aberrant expression patterns in mature hepatocytes, particularly under conditions of liver injury or disease[33], were observed.

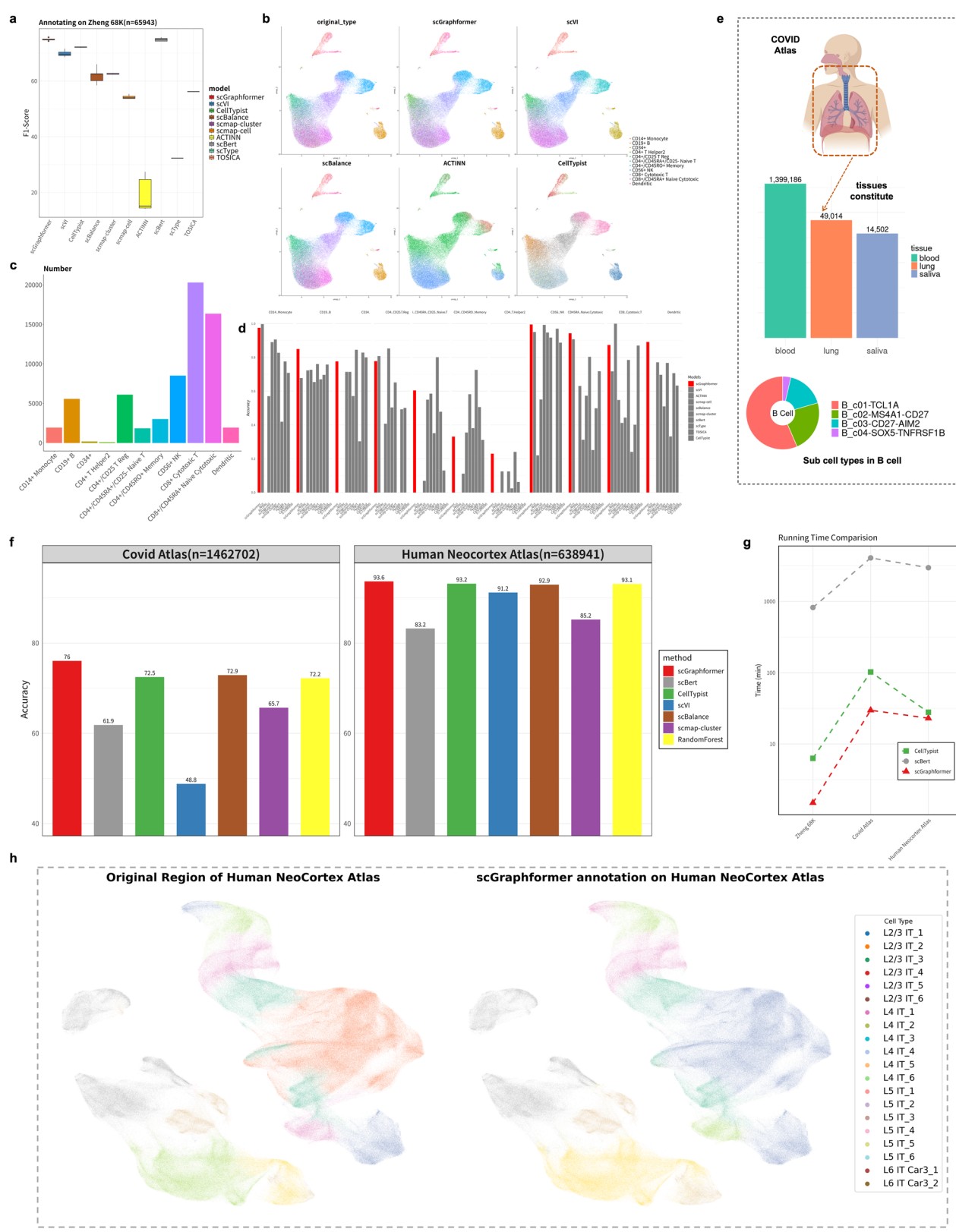

These patterns bear a resemblance to those found in immature hepatoblasts, as reflected in our results.

## Discussion
The advent of scRNA-seq technology has revolutionized our understanding of the intricate gene expression landscapes across heterogeneous cellular populations. Nonetheless, one of the enduring challenges in the field has been the analysis of sparse and high-dimensional expression data that typify scRNA-seq results. This study presents scGraphformer, an innovative approach that harnesses a graph transformer architecture to navigate the labyrinth of cell-cell topological relationships inherent to scRNA-seq datasets. Traditional GNN are predicated on message-passing algorithms

**Fig. 5 | scGraphformer's annotation performance on large-scale datasets reveals its scalability and identification capability on multiple cell types. a** Performance comparison of various cell type annotation methods on the Zheng 68K dataset. The *x*-axis shows different methods, while the *y*-axis represents the F1-score ranging from 30 to 80. Each boxplot summarizes the results of five experiments per method. **b** UMAP visualizations of cell type annotations across different methods in Zheng 68K dataset. The top-left panel shows the original true labels, serving as the ground truth. The subsequent panels display the annotation results from various methods. Each color represents a distinct cell type, as indicated in the legend. **c** The cell number of each cell type on Zheng 68K dataset and comparison of classification performance between scGraphformer and other methods. **d** This bar chart compares the performance of scGraphformer (red bars) against other annotation methods (gray bars) on the Zheng 68K dataset. Each group of bars represents a specific cell type, with performance metrics (likely accuracy or F1-score) on the *y*-axis ranging from 0 to 1.

**e** Overview of cell distribution in the Covid Atlas, displaying the diversity of 1.46 million cells across 64 cell types from blood, lung, and saliva tissues. The bar plot (lower left) quantifies cell numbers per tissue. We use sub-cell types as annotation labels and the pie charts illustrate the sub-cell type composition within B cells. **f** Classification accuracy comparison between scGraphformer and other methods on two human cell atlas (Covid atlas: 1.46 million cells, Human Neocortex atlas: 638K cells). The bar plot shows scGraphformer's annotation performance on two atlas compared to other methods. The number above each bar represents the accuracy score. And RandomForest is added as a traditional ML-based reference method. **g** Time-consuming comparison between two best models and LLM-based scBert. The *y*-axis uses minutes ranging from 0 to 10,000 min. The *x*-axis represents three large-scale datasets. **h** UMAP visualization shows annotation results on the human neocortex atlas, which reveals scGraphformer's superior performance in identifying minor cell populations even on large-scale datasets.

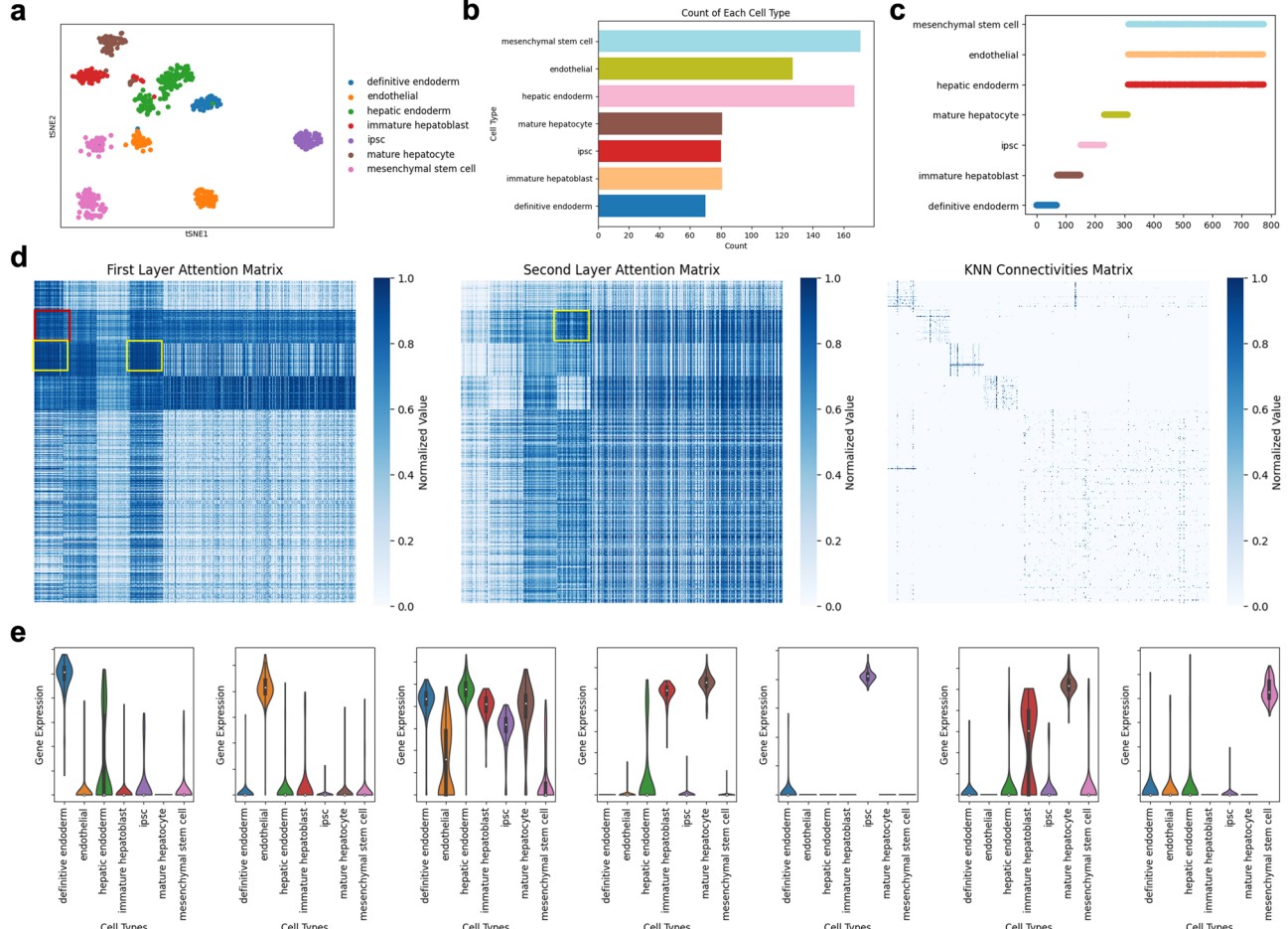

**Fig. 6 | Exploring cell interactions within the campLiver dataset using scGraphformer's attention mechanism. a** t-SNE visualization depicting clusters of various cell types. **b** Bar chart showing the count distribution of different cell types. **c** Scatter plot representing the distribution of cell types across cell indices. The numbers indicate some of the findings in the attention map. **d** Heatmaps showcase the attention matrices for the first (left) and second (middle) layers, alongside the kNN connectivity matrix (right). **e** Violin plots illustrate the expression levels of key marker genes across each cell type.

that incrementally exchange and process information across a network's edges. However, these algorithms often fall short of capturing the full spectrum of complex cell-cell interactions. ScGraphformer transcends these limitations by integrating a novel transformer computation mechanism with graph learning, thereby enabling a more holistic interrogation of cell-cell topologies. This comprehensive message-passing scheme allows for an enriched annotation of cells, preserving the nuanced spectrum of heterogeneity and homogeneity captured within transcriptomic profiles.

The scalability and performance of scGraphformer merit particular attention. As scRNA-seq datasets burgeon in size, the capacity to analyze large volumes of data without sacrificing accuracy becomes paramount.

scGraphformer distinguishes itself in this regard, offering whole-genome computation capabilities that not only mitigate batch effects but also enhance the interpretability of results. Our comparative analyses pitted scGraphformer against existing annotation methods and revealed its superior performance across various benchmarks. Notably, its proficiency in identifying rare cell populations within imbalanced datasets underscores the algorithm's sensitivity and robustness.

While scGraphformer marks a significant leap forward by marrying transformer and GNN architectures for cell type annotation, the conversion of genome-wide computational findings into actionable insights on cellular interactions poses a considerable hurdle. Addressing these challenges is the

thrust of our future work, which will aim to refine the algorithm's capacity to integrate gene regulatory networks and biological transcriptional signals, thereby enhancing its interpretative power.

In summary, the scGraphformer algorithm skillfully weaves together scRNA-seq data integration, graph structure learning, and the construction of a final all-pair cell-cell topology graph. This integrative approach enables the extraction of meaningful information from scRNA-seq data. Using the combined power of machine learning and graph network analysis, scGraphformer unveils hidden patterns and relationships within datasets. This facilitates a profound understanding of cellular heterogeneity and interactions.

## Methods

### Data processing and normalization

We represent each scRNA-seq dataset's gene expression matrix as $X \in \mathbb{R}^{v \times g}$, where $v$ is the number of cells and $g$ is the number of genes. Initially, we conduct data filtering and quality control, retaining only genes expressed in more than 1% of cells and cells with expression in more than 1% of genes due to the high dropout rate in gene expression profiles. The filtered matrix undergoes log transformation and normalization in accordance with the standard preprocessing protocol provided by Scanpy[34]. Subsequently, we identify and select highly variable genes, sorting them by normalized variance. The resulting log-normalized dataset ready for input is denoted as $\hat{X} \in \mathbb{R}^{v \times \hat{g}}$, where $\hat{g}$ represents the top-ranked genes.

### Cell graph construction

In addition to the expression matrix, the scGraphformer model can also work with an initial cell graph. Following standard practice, we construct this graph using the kNN algorithm, where graph nodes correspond to individual cells and edges indicate relationships among them. For each dataset, upon completing cell and gene filtering, we calculate the graph with a predefined k value, which determines the scale of interaction captured between cells. This k determines the number of nearest neighbors each cell node connects with, based on the Euclidean distance. Prior research has validated the efficacy of kNN in artificially modeling biological networks[35,36]. Ultimately, we define the input graph as $\mathcal{G} = (\mathcal{V}, \mathcal{E})$, where $\mathcal{V}$ denotes the set of cell nodes and $\mathcal{E}$ denotes the set of edges connecting to each cell. The graph structure is represented by a binary adjacency matrix $A \in \{0, 1\}^{|\mathcal{V}| \times |\mathcal{V}|}$. Combined with the above-normalized expression matrix, each cell node $u$ in the graph is associated with a feature vector $\hat{x}_u$ and a label $y_u$ and the overall feature matrix and class vector can be represented by $\hat{X}$ and $Y$, respectively.

### The structure of scGraphformer

The application of graph-based methods for cell annotation has been explored in previous efforts[14,37]. Nevertheless, the task of crafting graph transformer architectures that can match the growing scale of scRNA-seq data persists. A critical aspect of this challenge is designing a graph architecture that effectively integrates with the transformer structure and can capture interactions between cell nodes within the graph. This is especially critical when meeting scalability needs, requiring tailored global attention mechanisms to discern long-range interactions among cells. Implementing global attention, however, presents challenges when managing graphs with an indeterminate number of nodes. In this study, we address these issues by introducing scGraphformer.

For input cell expression $\hat{X} \in \mathbb{R}^{v \times \hat{g}}$, we first use a shallow fully connected layer to convert the gene expression into a $d$-dimensional cell embedding in the latent space:

$$Z = \sigma(LayerNorm(W_1 X + b_1))), \tag{1}$$

where $W_1 \in \mathbb{R}^{d \times \hat{g}}$ and $b_1 \in \mathbb{R}^d$ are trainable parameters of input layer, and $\sigma$ is a non-linear activation (i.e., ReLU). Then the new cell embeddings $Z$ will be used for feature propagation with our graph transformer model by letting $Z^{(0)} = Z$ as the initial states.

Subsequently, we refine the original attention computation in the Transformer to more effectively fulfill the requirements of learning within biological networks, where attention serves to capture cell interactions in the graph. This calculation is transposed into query, key, and value vectors, which are then aggregated to compute the attention across all cell node pairs:

$$q_u^{(k)} = W_Q z_u^{(k)}, \quad k_u^{(k)} = W_K z_u^{(k)}, \quad v_u^{(k)} = W_V z_u^{(k)}, \tag{2}$$

$$z_u^{(k+1)} = \sum_{v=1}^{N} \frac{exp((q_u^{(k)})^\top k_v^{(k)})}{\sum_{w=1}^{N} exp((q_u^{(k)})^\top k_w^{(k)})} v_v^{(w)}, \tag{3}$$

where we denote $z$ as the cell node embeddings, and $q$, $k$, $v$ as the vectors of query, key, and value, respectively, then we denote $W_Q$, $W_K$, $W_V$ as the learnable weights of three components.

In addition, to reduce the attention computation to graph learning results in a time complexity of $O(N^2)$ for updating $N$ cell nodes in one layer, as the computation of the aforementioned methods necessitates $O(N)$. The quadratic complexity is burdensome and may hinder scalability; thus, we aim to reduce the complexity from $O(N^2)$ to $O(N)$ while preserving the relationships between all cell node pairs. Consequently, we have incorporated a simplified attention computation to mitigate the complexity based on the Taylor expansion of the exponential function[38].

Thus, the approximated equation of attention can be written as,

$$
\begin{aligned}
z_u^{(k+1)} &= \sum_{j=1}^{N} \frac{1 + (\hat{q}_i^{(k)})^\top \hat{k}_j^{(k)}}{\sum_{l=1}^{N}(1 + (\hat{q}_i^{(k)})^\top \hat{k}_l^{(k)})} v_j^{(k)} \\
&= \frac{\sum_{j=1}^{N} v_j^{(k)} + (\sum_{j=1}^{N} \hat{k}_j^{(k)} (v_j^{(k)})^\top) \hat{q}_i^{(k)}}{N + (\hat{q}_i^{(k)})^\top \sum_{l=1}^{N} \hat{k}_l^{(k)}},
\end{aligned} \tag{4}
$$

This new attention computation layer could be efficiently computed using linear complexity where the computed weights are shared by all the nodes and thereby only need one-time computation. In original attention, the computation of $z_u^{(k+1)}$ is represented as $Softmax \times V^{(k)}$ algorithm when introducing each node embeddings to all cell nodes, and we introduce to all nodes here:

$$\hat{Q}^{(k)} = \frac{Q^{(k)}}{||Q^{(k)}||_2}, \quad \hat{K}^{(k)} = \frac{K^{(k)}}{||K^{(k)}||_2}, \tag{5}$$

$$Z^{(k+1)} = \frac{V^{(k)} + (\hat{K}^{(k)}(V^{(k)})^\top)\hat{Q}^{(k)}}{N + (\hat{Q}^{(k)})^\top \hat{K}^{(k)}} \tag{6}$$

Therefore, the transformer structure is re-designed for graph learning, especially for large-scale datasets.

Unlike GNN, which resorts to message passing over a fixed input graph topology, graph Transformers can more flexibly aggregate global information from all the nodes through adaptive topology in each propagation layer. When there is a high-quality cell graph, we can embed a GCN module in the scGraphformer to incorporate the structure information (this process is optional). Consider the input graph as $\mathcal{G}$, then we define the graph aggregation module as **GCN** whose target is to learn and update the cell graph topology structure, i.e., establishing edges between more homogeneous cell nodes and making sure cells from the same types connect. We thus establish the following rules:

$$\mathbf{GCN}^{(k)} \leftarrow Z^{(k)} + D^{-\frac{1}{2}} \mathcal{G} D^{-\frac{1}{2}} V^{(k)}, \tag{7}$$

and $D$ denotes its corresponding diagonal degree matrix, where $Z^{(k)}$ is calculated with equation (6). Thus, the next-layer embeddings will be updated

after the input graph structure add by:

$$Z^{(k+1)} = \sigma'(LayerNorm(\beta \cdot \mathbf{GCN}^{(k)} + (1-\beta)Z^{(k)})), \quad (8)$$

where $\sigma'$ can be identity mapping or non-linear activation where we choose ReLU as the activation function.

After K layers of propagation, we add a fully connected layer to output the predicted logits:

$$\hat{Y} = Z^{(K)}W + b, \quad (9)$$

where $W$ and $b$ are trainable parameters during the whole training. The predicted logits $\hat{Y}$ will be used for computing a loss of the form $l(\hat{Y}, Y)$ where $l$ denotes cross-entropy loss for cell annotation.

## Data description

To evaluate the efficacy of scGraphformer, we initially applied it to 20 diverse intra-datasets. These datasets were meticulously selected to encompass a broad range of sequencing platforms, species, organs, and tissues, and varied in cell count (including Adam[39], AMB[40], Bach[41], Baron Human[26], Baron Mouse[26], campbell[30], campLiver[23], darmanis[42], Deng[43], Klein[44], lake[45], Muraro[46], TM[47], Tosches turtle[48], usoskin[42], Segerstolpe[49], Xin[50], Young[51], Zheng 68K[25], and zillionis[24]). Each dataset was divided into training and testing subsets for the evaluation process. And we also use seven datasets using different protocols sampled from PBMCBench[27] to evaluate scGraphformer's performance in the inter-datasets experiments. Also, to evaluate our performance on complex and large datasets, we collected datasets such as Human Neocortex Atlas[52], Covid-19 immune Atlas[53], Zeisel[28] and Rosenberg[29]. And the Human Neocortex Atlas and Covid-19 Atlas were used for evaluating our scalable performance on a large amount number of cells. Zeisel and Rosenberg were used to perform the model's cross-platform ability on large data. Details of datasets can be found in Supplementary Table B1.

## Evaluation settings

We discussed our evaluation metrics for cell annotation performance and experiment settings for scGraphformer and comparison methods. We employed accuracy, F1-score and Cohen's Kappa Score to evaluate the performance of each method on cell type annotation in intra-experiments providing a more comprehensive evaluation of scGraphformer's capabilities (Supplementary Tables C2–C4). To conduct a more comprehensive evaluation of our cross-platform experiments, we introduced Normalized Mutual Information (NMI), Adjusted Rand Index (ARI), Balanced Accuracy, Precision and recall. These metrics allow us to thoroughly examine the performance of scGraphformer and other methods across different aspects of cell-type annotation tasks. All calculated using the scikit-learn package:

- Weighted F1-score: A harmonic mean of precision and recall, weighted by class support.

$$F1_{weighted} = \sum_{i=1}^{n} w_i \cdot F1_i \quad (10)$$

where $w_i$ is the support of class $i$.

- Accuracy score: The ratio of correct predictions to total predictions.

$$Accuracy = \frac{\text{Number of correct predictions}}{\text{Total number of predictions}} \quad (11)$$

- Cohen's Kappa Score: Measures inter-rater agreement for categorical items.

$$\kappa = \frac{p_o - p_e}{1 - p_e} \quad (12)$$

where $p_o$ is the empirical probability of agreement and $p_e$ is the expected probability of agreement.

- Normalized Mutual Information (NMI): Quantifies the mutual dependence between the predicted and true labels.

$$NMI(U, V) = \frac{2 \cdot I(U, V)}{H(U) + H(V)} \quad (13)$$

where $I(U, V)$ is the mutual information and $H(U)$, $H(V)$ are the entropies.

- Adjusted Rand Index (ARI): Measures the similarity between two clusterings.

$$ARI = \frac{\text{RI} - E[\text{RI}]}{\max(\text{RI}) - E[\text{RI}]} \quad (14)$$

where RI is the Rand Index.

- Balanced Accuracy: The average of recall obtained in each class.

$$BalancedAccuracy = \frac{1}{n} \sum_{i=1}^{n} \frac{TP_i}{TP_i + FN_i} \quad (15)$$

- Precision: The ratio of true positive predictions to total positive predictions.

$$Precision = \frac{TP}{TP + FP} \quad (16)$$

- Recall: The ratio of true positive predictions to all actual positive instances.

$$Recall = \frac{TP}{TP + FN} \quad (17)$$

We conducted each experiment five times with independent random splits of the datasets, maintaining the same proportions for input into the model. The results were recorded, and we calculated the standard error and mean accuracy to ensure the reliability and reproducibility of our findings. While scGraphformer can operate without a predefined graph, we experimented with incorporating a KNN-graph for each dataset during annotation. Contrary to our initial expectations, the addition of the KNN-graph did not significantly improve performance. This result suggests that scGraphformer's base architecture already captures essential cell relationships effectively (Details shown in Supplementary Fig. G5). The limited impact of the KNN-graph hints at the potential for more biologically relevant graph structures. We hypothesize that graphs based on known cellular interactions, developmental trajectories, or functional relationships might provide more meaningful improvements.

In intra-dataset experiments, we partitioned each dataset into training, validation, and testing subsets. For methods without a validation set in their original studies, we set the validation proportion to zero to compare performance fairly with scGraphformer. All hyperparameters were fine-tuned for optimal performance, and we maintained a train-validation-test split ratio of 0.6, 0.2, and 0.2 to ensure sufficient data for training and testing.

For cross-platform (inter-dataset) experiments, we used the reference dataset for training and the query dataset solely for testing. While training, the reference dataset is split into training data and valid data (0.8/0.2). We retained only the common genes between datasets to ensure a consistent input feature space. In fusion data experiment, for each evaluation of a single sequencing dataset, we integrated the remaining six datasets into a unified fusion dataset. Then, we trained the model in fusion dataset and tested on the excluded sequencing data.

For large-scale experiments (over 50,000 cells), we implemented mini-batch training to manage the high memory demands of topology structures and gene expression data. We divided the cell nodes into random mini-batches each epoch and focused learning on the topology structure within each batch. Even though this approach might reduce performance on all-pair cell interactions, scGraphformer still outperformed other methods. We set the batch size to 512, suitable for an NVIDIA 4090 GPU, and performed annotations for the testing set on a CPU to process the entire dataset.

Throughout these experiments, we reported test accuracy from the epoch with the highest accuracy on the testing dataset.

### Comparison methods and settings

In this study, we evaluated the performance of scGraphformer against seven popular scRNA-seq cell annotation methods: CellTypist[19], scVI[6], scmap-cluster and scmap-cell[9], ACTINN[8], scBalance[7], scBert[20], TOSICA[21], scType[22]. All methods were implemented according to the guidelines and tutorials provided by their respective packages. To ensure fairness in our comparisons, we subjected each method to a standardized preprocessing pipeline and maintained default parameter settings.

The running environment and version of comparison methods are as follows:(1) CellTypist from GitHub (https://github.com/Teichlab/celltypist) which we use pip to install the package. (2) scVI from GitHub (https://github.com/scverse/scvi-tools). (3) ACTINN from GitHub (https://github.com/mafeiyang/ACTINN), where we downloaded the whole project and only made data-preprocess modifications to adopt our dataset. (4) scBalance from GitHub (https://github.com/yuqcheng/scBalance), where we used pip to install the package. All of these methods use Python. (5) scmap using R language from GitHub (https://github.com/hemberg-lab/scmap). It includes two strategies: scmap_cluster and scmap_cell; scmap_cluster maps individual cells from query samples to certain cell types in the reference dataset, whereas scmap_cell maps individual cells from query samples to individual cells in a reference dataset. (6) scBert from GitHub (https://github.com/TencentAILabHealthcare/scBERT), where we processed datasets according to their requirements and predict cells based on their pre-trained weights. (7) TOSICA from GitHub (https://github.com/JackieHanLab/TOSICA). (8) scType from GitHub (https://github.com/IanevskiAleksandr/sc-type).

All of the methods are using their default parameters and settings. We only made data preprocess modifications to match our experiment datasets. All of the experiments were running on the workstation with Intel(R) Xeon(R) W-2223 CPU @ 3.60 GHz, Ubuntu 22.04, NVIDIA 4090 GPU and 64GB RAM, and each method's environment follows the Python version.

### Statistics and reproducibility

Data manipulation and processing analyses were conducted using the packages Python (version 3.8), R (version 4.4.1), Pandas (version 2.0.3), numpy (version 1.24.3), scipy (version 1.10.1), and PyTorch (version 1.13.1). We used scanpy (version 1.9.6), anndata (version 0.9.2) and scvi-tools (version 0.20.3) to preprocess the scRNA-seq dataset. We used torch_geometric package (version 2.4.0) to construct and process the cellular graph. The plots in our study are drawn by Matplotlib (version 3.4.3), Seaborn (version 0.12.2) and ggplot2 (version 3.5.1).

### Reporting summary

Further information on research design is available in the Nature Portfolio Reporting Summary linked to this article.

### Data availability

No new data were generated for this study. All data used in this study are publicly available, as previously described. And we have uploaded all processed data into the cloud (https://mycuhk-my.sharepoint.com/:f:/g/personal/1155187720_link_cuhk_edu_hk/EqVfLiFZDApEtel_fLOX_8gBCC83cvuz7o4UgZrAfEtFyw).

### Code availability

We have opened the GitHub repository of scGraphformer (https://github.com/xyfan22/scGraphformer). The numerical source data for graphs and charts in this article is provided in Figshare with DOI[54]. We also deposit codes of scGraphformer in Figshare with DOI[55].

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

## Acknowledgements

The work described in this paper was supported partially by a grant from the Research Grants Council of the Hong Kong Special Administrative Region, China (Project Reference Number: T45-401/22-N) and by a grant from the Hong Kong Innovation and Technology Fund (Project Reference Number: ITS/241/21).

## Author contributions

Pheng-Ann Heng, Guangyong Chen, and Jiacheng Liu conceived the research topic. Xingyu Fan designed and developed the method and conducted the computational benchmark experiments. Guangyong Chen, Jiacheng Liu, Bian Wu, and Yirong Jiang provided the necessary domain background and assisted with experiments. Xingyu Fan, Guangyong Chen, and Jiacheng Liu designed and performed the case experiments. Chunbin Gu, Yuqiang Han, and Bian Wu contributed to the computational analysis. Xingyu Fan, Guangyong Chen, Jiacheng Liu, Yaodong Yang, and Yuqiang Han participated in writing the paper. Guangyong Chen, Jiacheng Liu, and Pheng-Ann Heng supervised the work. All co-authors participated in discussions and agreed with the contents of this work.

## Competing interests

The authors declare no competing interests.
