## [Transparent Peer Review file · Communications Biology]

scGraphformer: Unveiling Cellular Heterogeneity and Interactions in scRNA-seq Data using a Scalable Graph Transformer Network

Corresponding Author: Professor Guangyong Chen

Version 0:

Reviewer comments:

Reviewer #1

(Remarks to the Author)

The authors developed a new method called scGraphformer, which combines transformer and graph neural network to capture cell-cell interactions and can be applied to cell type identification tasks. Results presented in the article demonstrate that scGraphformer outperforms several other methods in cell type annotation, particularly on large-scale datasets and when identifying rare cell types within imbalanced datasets. However, several issues need to be addressed:

1. In Fig. 2b, some methods including scmap-cluster, scmap-cell, ACTINN, and ScBalance lack results on the Zheng 68k dataset. Please explain the reason. Additionally, Table B1 shows that the Baron Human dataset has 14 cell types, yet Fig. 2c only displays results for 13 cell types. The colors representing accuracy levels of 90, 95, and 100 in Fig. 2a are indistinguishable; it's recommended to replace them with more distinguishable colors.
2. The compared methods in Fig. 2 are not very novel; for instance, scVI and scmap were developed in 2018. Nowadays, more cell type annotation methods, especially those based on transformers like scBERT [1], scGPT [2], CIForm [3], TOSICA [4], scTransSort [5], TransCluster [6], are suggested to be considered. Marker gene-based annotation methods like ScType [7], GPT-4 [8] are also suggested to be included in the comparison.
3. In Fig. 3c, results for scBalance, scVI, and CellTypist appear competitive with scGraphformer. It's also challenging to discern where scGraphformer outperforms other methods in Fig. 3d, except for ACTINN.
4. In Fig. 3a and 3b, authors trained the model separately on 10Xv2 and 10Xv3 as training data and used other datasets as query datasets. How about training the model on multiple datasets or even all datasets?
5. Mentioned in Line 252, Fig. 4b should correspond to fig. 4a. Fig. 4b lacks corresponding content in the main text and its statistical information isn't necessary; it is suggested to be deleted.
6. From the confusion matrix in Fig. 4d, scmap-cluster performs better in identifying macrophage cells with a lower proportion in the dataset than scGraphformer. scmap-cell was compared in Fig. 2 and Fig. 3 but not in Fig. 4d; please provide scmap-cell's results or explain its exclusion.
7. It's unclear where scGraphformer is superior to RandomForest and CellTypist in Fig. 5b. Quantitative assessment will be helpful.
8. Line 286 mentions comparing scGraphformer with scBalance, yet scBalance's results aren't shown in Fig. 5b. Why are the methods compared in Fig 5b and Fig 5d inconsistent and different from those in Figs 2, 3 and 4?
9. scGraphformer's performance is limited on large-scale datasets, with significantly lower accuracy compared to other datasets, as shown in Figure 2b. This is doubtful about the performance if compared to large language model-based methods specifically designed for large-scale datasets.
10. In Supplementary Fig.E2, the colors for each cell type aren't unique, causing confusion.
11. The results of Fig. 5e are confusing. Intuitively, the annotation results of two classes are obviously inconsistent with ground truth, but Fig.2d shows that scGraphformer has an accuracy of more than 0.93. The model was trained using subtype labels, but how is the accuracy calculated? Did the authors calculate the accuracy of 64 sub cell types or the accuracy of 12 major cell types? Should the calculation of accuracy be consistent with the labels used for training?
12. In the Methods of "The structure of scGraphformer", authors said "When there is a high-quality cell graph, we can embed a GCN module in the scGraphformer to incorporate the structure information." From my understanding, if no prior cell graph is known, cell embedding z is updated according to equation (6) via Q,K,V transformation. However, if a high-quality cell graph exists, cell embedding z is updated according to equation (8), which has nothing with the attention mechanism. Is my understanding correct? It's better to make it more clear in the methods. For my curiosity, how to evaluate whether a cell graph is of high quality?
13. Overall, cell type annotation for scRNA-seq is a well-studied topic, the significance is somehow limited, but the method is of some novelty.

References

1. Yang, F., et al., scBERT as a large-scale pretrained deep language model for cell type annotation of single-cell RNA-seq data. *Nature Machine Intelligence*, 2022. 4(10): p. 852-866.
2. Cui, H., et al., scGPT: toward building a foundation model for single-cell multi-omics using generative AI. *Nat Methods*, 2024.
3. Xu, J., et al., ClForm as a Transformer-based model for cell-type annotation of large-scale single-cell RNA-seq data. *Brief Bioinform*, 2023. 24(4).
4. Chen, J., et al., Transformer for one stop interpretable cell type annotation. *Nat Commun*, 2023. 14(1): p. 223.
5. Jiao, L., et al., scTransSort: Transformers for Intelligent Annotation of Cell Types by Gene Embeddings. *Biomolecules*, 2023. 13(4).
6. Song, T., et al., TransCluster: A Cell-Type Identification Method for single-cell RNA-Seq data using deep learning based on transformer. *Front Genet*, 2022. 13: p. 1038919.
7. Ianevski, A., A.K. Giri, and T. Aittokallio, Fully-automated and ultra-fast cell-type identification using specific marker combinations from single-cell transcriptomic data. *Nat Commun*, 2022. 13(1): p. 1246.
8. Hou, W. and Z. Ji, Assessing GPT-4 for cell type annotation in single-cell RNA-seq analysis. *Nat Methods*, 2024.

Reviewer #2

(Remarks to the Author)

Please see the attachment.

Version 1:

Reviewer comments:

Reviewer #1

(Remarks to the Author)

The authors fully addressed my questions in the revision and incorporated comparisons with several newer methods, including large language model approaches, demonstrating the superior performance of scGraphformer in cell type annotation. The authors revised the article very seriously and supplemented the comparison of methods, and the article has strong readability.

Reviewer #2

(Remarks to the Author)

I have not further comments.

Dear Reviewers,

We greatly thank you for your constructive comments on this submission. These opinions helped us improve our paper significantly. Based on your suggestions, we have made corrected modifications to the manuscript. Please find below the point-to-point response to your comments.

Sincerely,
All authors.

Black part – Reviewers' Comments

Blue part – Answering

Red part -original manuscript

Orange part – Corresponding revision in the manuscript

***Reviewer #1:***

**Comment:**

The authors developed a new method called scGraphformer, which combines transformer
and graph neural network to capture cell-cell interactions and can be applied to cell type
identification tasks. Results presented in the article demonstrate that scGraphformer
outperforms several other methods in cell type annotation, particularly on large-scale
datasets and when identifying rare cell types within imbalanced datasets. However, several
issues need to be addressed.

**Answer:**

Thank you for your thorough and constructive review of our paper! We truly appreciate the
time you took to delve into the details. As you mentioned, our method captures cell-cell
interactions using a graph neural network and is applicable to cell type identification. We also
assessed our model's performance on large-scale datasets, particularly in the context of
imbalanced datasets containing rare cell types.

We have carefully considered all your suggestions. In this revision, we have made further
improvements to the manuscript based on your feedback and re-evaluated all experiments
after recognizing previous consistency issues. Thank you once again for your valuable insights!

**Comment 1:**

In Fig. 2b, some methods including scmap-cluster, scmap-cell, ACTINN, and ScBalance lack
results on the Zheng 68k dataset. Please explain the reason. Additionally, Table B1 shows that
the Baron Human dataset has 14 cell types, yet Fig. 2c only displays results for 13 cell types.
The colors representing accuracy levels of 90, 95, and 100 in Fig. 2a are indistinguishable; it's
recommended to replace them with more distinguishable colors.

**Answer 1:**

Thank you so much for your comments and mentioning the corresponding problems. We
sincerely appreciate your careful examination of our figures and tables.

Thank you for your observation. Regarding Fig. 2b, the accuracy gradient ranges from 70% to
100%. The evaluation performance of scmap-cluster, scmap-cell, ACTINN, and scBalance on
the Zheng 68k dataset is below 70%, which is why their results are not displayed in this figure.
However, their performance is reported in Supplementary Table C2, with scmap-cluster at
$59.13 \pm 0\%$, scmap-cell at $54.37 \pm 0\%$, ACTINN at $41.01 \pm 7.56\%$, and scBalance at $61.62 \pm 3.53\%$.
We appreciate you bringing this to our attention. To address this issue and improve clarity,
we have modified Fig. 2b by adding a dashed box within the Zheng 68K dataset column. This
dashed box indicates that the performance of these methods on this dataset is below the
lower boundary of the displayed accuracy range. The updated part in Fig. 2b is shown below.
And we extended this annotation to all datasets in our study. To visually represent the
performance and applicability of each method across different datasets, we implemented a
graphical notation system. Solid frames indicate that the method's performance on the
respective dataset is below 70%, while dashed frames signify that the method is not applicable
to that particular dataset. This visual approach allows for a quick and intuitive understanding
of each method's strengths and limitations across various data contexts.

Figure. 1: Modification on Fig. 2b to explain the empty part

After that, thank you for your astute observation regarding the Baron Human dataset. We
discovered that our initial process inadvertently filtered out cell types with fewer than 10 cells,
excluding the 't_cell' type (7 cells). Recognizing this oversight, we have re-evaluated our
performance, including all cell types regardless of size. We've updated Figure 2c to reflect all
14 cell types in the Baron Human dataset, adding cell count information for each type. Figure
2d has also been modified to match these results and improve clarity. Also, updated results
are now included in Table C4. These changes provide a more accurate representation of our
method's performance on imbalanced datasets. We appreciate your careful review, which has
helped us enhance the accuracy and completeness of our results.

Figure. 2: Modification on Fig. 2c

Figure. 3: Modification on Fig. 2d

Then, we appreciate your concern and have made significant modifications to Figure 2a based
 on your valuable suggestion. To address the issue of color distinguishability, we have
 implemented a more nuanced color scheme that clearly differentiates between various
 accuracy levels, especially in the high-performance range (90-100%). This new color palette
 allows for easier distinction between accuracy levels of 90%, 95%, and 100%. Additionally, we
 have refined the overall layout of the figure. The heatmap now uses a broader range of colors
 to represent different accuracy levels more clearly. We have also adjusted the boxplot to be
 more compact and integrated it seamlessly with the heatmap. Furthermore, we modified the
 proportion between the heatmap and the boxplot to create a more balanced and informative
 visual representation. These changes may enhance the overall clarity and readability of Figure
 2a, making it easier for readers to interpret our model's performance across various datasets.

Figure 4: Modification on Figure 2a

 We greatly appreciate your insightful comments, which have been instrumental in improving
 our manuscript. In response, we have made comprehensive revisions to the relevant sections
 of our paper. The figures presented above incorporate feedback from all reviewers, including
 your valuable input. We believe these changes could enhance the clarity and quality of our
 work, and we thank you for your contribution to this improvement process. And we have
 revised the caption of Fig. 2 in our manuscript, shown below.

 **Manuscript**

**Section “scGraphformer provides better performance than other single-cell**
 **classification methods in intra-dataset”**

Caption of Fig .2:

“scGraphformer shows its impressive performance in annotating cell types in intra-dataset. (a)
 Heatmap portraying the accuracy of five replicates for each scRNA-seq dataset, where color
 depth corresponds to accuracy level. The adjacent boxplot illustrates the distribution of
 accuracies per dataset, delineating the median (central line), interquartile range (box), and
 any outliers (dots). The number within each box represent the accuracy of annotation in each
 dataset. (b) A comparative assessment of scGraphformer and alternative methods within

intra-datasets, each subjected to quintuplicate evaluations to confirm consistency. Boxplots
illustrate scGraphformer's comparative annotation performance, revealing its consistently
higher median accuracy and reduced variability, particularly in complex datasets like Zheng
68K, TM, and zillionis. The solid frames indicate that the method's performance on the
respective dataset is below 70%, while dashed frames signify that the method is not applicable
to that particular dataset. (c) A cell-type-specific precision analysis within the Baron Human
dataset, demonstrating scGraphformer's heightened accuracy in identifying scarce cell types
relative to competing approaches. (d) Sankey diagram representing scGraphformer's
annotation accuracy within the Baron Human dataset, showcasing the correspondence
between predicted and true cell type classifications."

**Comment 2:**

The compared methods in Fig. 2 are not very novel; for instance, scVI and scmap were
developed in 2018. Nowadays, more cell type annotation methods, especially those based on
transformers like scBERT [1], scGPT [2], ClForm [3], TOSICA [4], scTransSort [5], TransCluster
[6], are suggested to be considered. Marker gene-based annotation methods like ScType [7],
GPT-4 [8] are also suggested to be included in the comparison.

**Answer 2:**

We appreciate your insightful comments. Following your suggestions, we have expanded our
comparison to include several state-of-the-art cell type annotation methods. Specifically, we
have incorporated **scBERT** to represent the latest advancements in LLM-based approaches,
**TOSICA** for transformer-based methods, and **scType** for marker gene-based approaches.
These updates are reflected in the new Fig. 2b in the paper, and additional details are
illustrated in Figure 5 below. For your reference in the figures, the black dashed box indicates
that the dataset is not suitable for the method in question. Additionally, the red box signifies
that the performance was below 70%, leading to its exclusion from the figure. The revised
result is shown in below Figure. 5 which correspond to our Fig.2b in our manuscript.

In our revised analysis, we identified that some datasets are not compatible with the
aforementioned methods, as indicated by black dashed boxes in our figures. For instance,
scBERT relies on the inclusion of specific genes from its pretraining dataset, Panglao DB, in
the fine-tuning dataset. However, some of our datasets, such as Campbell and Adam, do not
contain these genes, which limits scBert's ability to accurately annotate cell types. To ensure
a comprehensive evaluation, we assessed its performance under both zero-shot and fine-
tuning scenarios which demonstrated scBert hardly achieve a good performance without
seeing any information of the annotated datasets. The method's performance was
constrained by its limited domain-specific knowledge of genes, which significantly impacted
its overall effectiveness. Nevertheless, we remain intrigued by the potential of LLM-based
approaches in this field, as they have demonstrated remarkable adaptability in learning
complex transcriptomic patterns by self-supervised learning paradigm.

TOSICA was found to be unsuitable for some datasets as it only supports gene ontology
biological processes (GOBP) for humans and mice, thus precluding its use with datasets
derived from other species. In the case of scType, we encountered discrepancies between its

annotated cell types and the reference cell types. For example, in the Zheng 68K dataset, we
 manually adjusted the cell type annotated by scType from 'effector cd8+ t' to 'cd8+ cytotoxic
 t' to align with the original dataset. All datasets with significant discrepancies were rigorously
 reviewed and those with irreconcilable differences were excluded, hence the dashed boxes.
 To make fair evaluation, all of results have been manually checked by us to ensure the
 consistency.

Furthermore, our evaluation revealed that models requiring extensive prior knowledge
 generally underperformed in our settings, likely due to their specialized design for specific
 tasks. We also explored the recently popular chat-marker-based identification model,
 selecting GPTCellType as our evaluation model. We assessed GPTCellType using both 'gpt-
 3.5' and 'gpt-4' versions. However, we found the performance unsatisfactory as GPTCellType
 primarily provided general identification based on the provided marker genes (where we set
 the top gene number to 20/30/50). So, we didn't include these result into our manuscript or
 supplementary material. But GPT-based model also shows its potential ability on cell
 annotation in scRNA-seq.

In conclusion, the sustained superior performance of scGraphformer, even after the inclusion
 of other advanced models, underscores its effectiveness and adaptability. The challenges
 encountered with the newly added models on our datasets highlight the need for continued
 research into developing more universally adaptable and robust cell type annotation methods
 that can deliver consistent performance across diverse biological datasets. And we also
 revised the content in our manuscript in section "scGraphformer provides better performance
 than other single-cell classification methods in intra-dataset" shown below.

Figure. 5: Modification on Fig.2b

**Manuscript**

**Section “scGraphformer provides better performance than other single-cell**
**classification methods in intra-dataset.”**

First Paragraph

“To evaluate the performance of scGraphformer, a series of intra-dataset evaluations were
conducted across 20 diverse datasets (Supplementary Table B1). The classification capabilities
of scGraphformer were benchmarked against seven state-of-the-art computational
strategies commonly utilized in scRNA-seq cell annotation: CellTypist, scVI, scmap-cluster,
scmap-cell, ACTINN, scBert, TOSICA, scType and scBalance.”

Second Paragraph

“Some methods also show outstanding performance on several datasets but some of them
are not sensitive to minor cell types and struggle to identify rare populations. Meanwhile,
several methods faced limitations in our evaluation. For example, scBERT's performance was
hindered by the absence of specific genes in some datasets. TOSICA's applicability was
restricted due to its support for only human and mouse GOBP. scType's annotations
sometimes mismatched reference cell types, requiring manual adjustments. These constraints
highlight the challenges in developing universally applicable cell type annotation tools. And
we use dashed box to represent their absence in that dataset.”

**Comment 3:**

In Fig. 3c, results for scBalance, scVI, and CellTypist appear competitive with scGraphformer.
It's also challenging to discern where scGraphformer outperforms other methods in Fig. 3d,
except for ACTINN.

**Answer 3:**

Thank you for your insightful observation regarding Fig. 3c. We acknowledge that the
performance of scGraphformer appears comparable to scBalance, scVI, and CellTypist in
certain aspects, as shown in Figure 3c. This indeed demonstrates the competitive landscape
of current cell type annotation methods. And we think the results may be more convincing if
we add more evaluation metrics to compare scGraphformer and other methods.

To address this, we have conducted an extensive re-evaluation of all methods across 42
cross-platform datasets, incorporating a broader range of evaluation metrics. These
additional metrics include Accuracy, Adjusted Rand Index (ARI), Balanced Accuracy, Cohen's
Kappa Score, F1-Score, Normalized Mutual Information (NMI), Precision, and Recall. The
comprehensive results have been added to our supplementary material for a more thorough
comparison. The attach Figure. 6 presents a summary of this expanded analysis, showing the
average performance and standard deviation for each method across all 42 datasets for each
metric. And the mean and median value of scGraphformer which surpassed other methods
exhibit its stability (Table 1). This multi-dimensional evaluation offers a more nuanced view of
the relative strengths of each method: 1. Consistency: scGraphformer demonstrates
consistent performance across all metrics, indicating its robustness and reliability across
diverse datasets and evaluation criteria. 2. Competitive edge: While scGraphformer may not
outperform all other methods in every single experiment, it shows competitive or superior

results in several key metrics. For instance, it performs particularly well in Accuracy, ARI, and
 Cohen's Kappa Score, which are crucial indicators of overall annotation quality and agreement.
 3. Balanced performance: The results suggest that scGraphformer achieves a good balance
 across different performance aspects, without sacrificing one metric for another. This is
 evident in its strong showing across precision, recall, and F1-score simultaneously. 4. Stability:
 The relatively small error bars for scGraphformer across most metrics indicate consistent
 performance across different datasets, suggesting good generalizability. We have uploaded
 results under those metrics into our supplementary material.

While we acknowledge that other methods like scBalance, scVI, and CellTypist also show
 strengths in certain areas, this comprehensive evaluation demonstrates that scGraphformer
 offers a robust and competitive solution for cell type annotation. Its balanced performance
 across multiple metrics and datasets underscores its value as a versatile tool in the field of
 single-cell analysis. We believe this expanded analysis provides a more complete picture of
 scGraphformer's performance in the context of other state-of-the-art methods, addressing
 the nuances that may not have been fully captured in the original figures 3c and 3d.

And all the re-evaluated results have been added to our revised supplementary material
 (Supplementary Appendix D). We also changed Figure 5 to a 4-row, 2-column shape and
 added it to Figure3 in the article. Notably, the methods added upon recommendation (scBert,
 TOSICA, and scType) were not used in this part of the study, as they are not supervised
 methods and our experimental setup required supervised learning for cross-platform
 evaluation.

Table 1: Summary reports on all cross-platforms experiments where we calculate the median,
 mean values under all evaluation metrics.(Supplementary Table D10)

method	Accuracy			F1-Score			NMI			Cohen's Kappa Score		
	median	mean	std	median	mean	std	median	mean	std	median	mean	std
scGraphformer	94.92	94.01	2.31	94.29	92.83	3.48	86.8	86.71	2.24	93.05	91.77	2.98
scBalance	94.13	93.28	2.19	93.96	92.43	3.42	85.84	85.75	1.97	92.37	91.25	2.84
scVI	92.69	92.4	2.52	92.94	91.35	3.41	86.02	85.86	2.62	90.38	90.04	3.23
CellTypist	93.42	92.26	2.26	90.6	90.31	3.61	83.47	83.03	1.9	91.29	89.88	2.92
scmap-cell	89.61	89.41	1.73	90.07	89	3.01	78.04	77.86	1.58	86.63	86.28	2.23
scmap-cluster	82.92	83.28	1.81	83.78	83.57	2.44	69.64	69.76	1.57	78.27	78.56	2.17
ACTINN	80.69	80.61	2.64	76.69	76.74	3.74	69.21	69.64	1.93	74.46	73.95	3.48

Figure. 6: Results comparison across 42 experiments using 8 metrics.

**Manuscript**

**Content in Section "scGraphformer is robust in inter-dataset evaluation"**

Caption of Fig. 3:

"Comparison of scGraphformer performance across different scRNA-seq protocols. (a)
Performance of scGraphformer when trained on 10Xv2 protocols, and applied to annotate
cell types from other protocols. (b) Comparative performance of scGraphformer trained on
single-platform data versus fusion data across various cross-platform scenarios. The x-axis
represents different query datasets, while the y-axis shows the F1-Score, ranging from 85 to
100. Box plots depict the statistical distribution of results for each dataset and data type.
Green boxes represent models trained on fusion data, which integrates data from all
sequencing technologies except the one being queried. Orange boxes represent models
trained on single-platform data. (c) Overall annotation ARI scores of scGraphformer in
comparison to other methods, with each train-test experiment pair labeled as "Train
Dataset\Test Dataset." Accuracy scores serve as the primary metric of evaluation, with 42
distinct train-test combinations illustrated. The accompanying boxplot provides a summary
of the comparative performance. The results shows scGraphformer performs better in the
overall ARI score. (d) comprehensive comparison of performance metrics across all cross-
platform experiments for different methods. Eight metrics are displayed: Accuracy, ARI,
Balanced Accuracy, Cohen's Kappa Score, F1-score, NMI, Precision, and Recall. Each subplot
uses box plots to show how the seven methods (scGraphformer in blue, others in gray)
perform across all experiments. scGraphformer consistently outperforms other methods in
most metrics, especially in Accuracy, ARI, F1-score, and NMI, showing both higher median
values and less variation. (e) UMAP visualization using scanpy displaying the classification of
cell populations within the 10Xv3 dataset, annotated according to predictions from various
methods, all of which were initially trained using the Smart-Seq PBMC dataset. scGraphformer
shows a particular strength in the precise identification of rare cell types, as indicated by the
distinct and accurate annotation of clusters. ~~The model outperforms other methods in~~
~~classifying challenging minor populations. Additionally, scGraphformer demonstrates~~
~~adaptability by assigning previously unclassified cells, such as megakaryocytes, to the~~
~~dendritic cell category based on shared immune function gene expression profiles, which are~~
~~prominent in the Smart-Seq training data. This highlights scGraphformer's potential to~~
~~adaptively recognize and reconcile functional similarities between cell types."~~

Second Paragraph

"Preliminary results, illustrated in Fig. 2a and Fig. 2b, demonstrate scGraphformer's proficiency
in maintaining annotation accuracy across datasets when ~~respectively~~ trained on 10Xv2 and
~~10Xv3~~ and tested on other platforms; and when trained on 10Xv2 and tested on Seq-Well,
scGraphformer achieved a mean accuracy of 95.46% ~~96.08%~~, which is approximately 2% ~~3.54%~~
higher than the average of ~~other~~ the second best methods (CellTypist: 93.66% , details shown
in Supplementary Table D57). During training, genes that were not expressed in both training
and testing data were excluded from the feature space, thereby enhancing the method's
resilience to batch-induced variability. The results indicate that scGraphformer's ability to
overcome batch effect. Furthermore, to compare its performance with other methods under
batch effect, we evaluate the annotation performance of all pairwise train-test combinations

between 7 protocols on the left methods. During training, genes that were not expressed in
both training and testing data were excluded from the feature space, thereby enhancing the
method's resilience to batch-induced variability. The results indicate that scGraphformer's
ability to overcome batch effect. Furthermore, to compare its performance with other
methods under batch effect, we evaluate the annotation performance of all pairwise train-
test combinations between 7 protocols on the left methods. All 49 experiment results are
summarized in Fig. 2c and the ~~accuracy score~~ Adjusted Rand Index (ARI) score is used as the
evaluation metric. And we set training proportion of 80% in reference datasets for realistic
situations (details in Supplementary Appendix D and Table D8). Besides ARI-score, we also
used accuracy score, Normalized Mutual Information (NMI), F1-score to compute the
annotation performance (Supplementary Fig D2, Fig D3 and Table D7). Under each metric,
scGraphformer achieves better in overall performance which has better median and mean
value across all experiments. (Supplementary Table D10 and Fig. 2d). ~~The percentage~~
~~improvement of scGraphformer over other methods ranges from a modest 0.08% to as high~~
~~as 5.18%, depending on the specific reference-query pair (Supplementary Table D6).~~ This
variance underscores the method's adaptability to diverse dataset characteristics and
sequencing platforms. The results show that scGraphformer achieved a good performance
across all experiments and there is not much difference from other state-of-art methods.
Moreover, scGraphformer's performance at rare cell type identification was also
demonstrated through cross-platform experiments, where we employed Uniform Manifold
Approximation and Projection (UMAP) for visualization purposes while training on Smart-seq
and testing on 10Xv3 (Fig. 2e and Supplementary Table D9), and it visualizes the clustering
result of scGraphformer with the original true cell types and predicted cell types. This aspect
of the analysis highlighted scGraphformer's superior performance in accurately annotating
minor cell populations, a critical capability given the frequent occurrence of imbalanced
datasets in scRNA-seq studies.”

**Supplementary Appendix D.**

First Paragraph

“And we compute their F1-score, NMI and ARI score where we computed their mean value
and its standard deviation according to their results. And the Fig.D2 and Fig.D3 shows their
results. We also put the results of scGraphformer under accuracy metric on the 42
experiments and each five results for each experiment (Table. D7).”

“Therefore, we choose the 0.8-proportion as our results on the paper since the demands of
realistic situation. Table \ref{tab:CR_SMART_10Xv3} shows the results of each method while
the training on SMART-seq2 and using the trained model to annotate 10Xv3. And the results
correspond to the UMAP visualization in Figure 2e in manuscript. For summarizing results on
all experiments, we computed the mean, median and standard variation values of
scGraphformer and other comparison methods across all 42 experiments which is shown in
Table \ref{tab:CR_average}.”

**Comment 4:**

In Fig. 3a and 3b, authors trained the model separately on 10Xv2 and 10Xv3 as training data
and used other datasets as query datasets. How about training the model on multiple datasets

or even all datasets?

**Answer 4:**

We appreciate the reviewer's insightful suggestion regarding the training of our model on
multiple datasets. In response to your suggestion, we retrained our model using multiple
datasets, excluding the query datasets designated for testing. Specifically, in our cross-
platform experiments, we evaluated our model on PBMC datasets sampled using seven
different technologies: Seq-Well, inDrop-seq, CEL-Seq, 10Xv2, Drop-Seq, Smart-Seq2, and
10Xv3.

To clarify, when evaluating one of the sequencing datasets, we integrated the remaining six
datasets into a single fusion dataset. This integration was done in such a way that no
genomic information was lost; genes not present in the original data were represented with
zero expression in the fused dataset. This process resulted in seven distinct fused datasets.
Each fused dataset was then used as a reference, while the dataset excluded from the fusion
served as the query dataset. The model was trained on the reference dataset and
subsequently used to annotate the query dataset.

The performance of our methods was assessed using the F1-score, which considers both
precision and recall, making it particularly suitable for cross-platform experiments. The
results of these experiments are displayed in the new plots included in our submission (as
shown below).

As depicted in the figure below, training on fusion data generally enhances the model's
performance compared to training on single datasets. This improvement underscores the
benefit of leveraging diverse datasets for model training in cross-platform scenarios. Here
we compare the F1-scores obtained when training on fused datasets versus single datasets
across various sequencing technologies. Green boxes represent the performance on fused
data, and orange boxes represent average performance across all reference datasets while
we use the other six dataset for reference. The chart illustrates that models trained on fused
data tend to perform better, showcasing higher median F1-scores and less variability
compared to those trained on single datasets.

Figure. 7: The figure compares the F1-scores obtained when training on fused datasets versus
single datasets across various sequencing technologies.

And we added the fused comparison in to our Fig2 which replace the original Fig2b.
Meanwhile, we have also added new contexts in our manuscript. Thank you so much for your
insightful comment. The revised contexts are also updated as follows, locating in Appendix D
in Supplementary Material

**Manuscript**

**Content in Section “scGraphformer is robust in inter-dataset evaluation”**

Caption of Fig. 3:

“(b) Comparative performance of scGraphformer trained on single-platform data versus
fusion data across various cross-platform scenarios. The x-axis represents different query
datasets, while the y-axis shows the F1-Score, ranging from 85 to 100. Box plots depict the
statistical distribution of results for each dataset and data type. Green boxes represent models
trained on fusion data, which integrates data from all sequencing technologies except the
one being queried. Orange boxes represent models trained on single-platform data.”

Third Paragraph

“Meanwhile, we explored the performance of training on fused datasets to simulate real-
world scenarios where biologists often use mixed data for annotating newly sequenced
datasets. For each evaluation of a single sequencing dataset, we integrated the remaining six
datasets into a unified fusion dataset. This approach aligns with practical applications where
datasets are not strictly separated by sequencing technology. Fig. 2b illustrates the
comparison between training on fusion data versus single data across different sequencing
platforms. The results consistently demonstrate that training on fusion data outperforms
single data training across all platforms. These findings validate our hypothesis that mixed
data can enhance annotation performance and contribute to more accurate cellular
identification. Detailed results and statistical analyses are provided in Supplementary
Appendix D, Tables D11 and D12.”

**Content in Methods “Evaluation settings”**

Fourth Paragraph

“For cross-platform (inter-dataset) experiments, we used the reference dataset for training
and the query dataset solely for testing. While training, the reference dataset is split into
training data and valid data (0.8/0.2). We retained only the common genes between datasets
to ensure a consistent input feature space. In fusion data experiment, for each evaluation of
a single sequencing dataset, we integrated the remaining six datasets into a unified fusion
dataset. Then we train the model in fusion dataset and tested on the excluded sequencing
data.”

**Supplementary Appendix D.**

Second Paragraph

“Meanwhile, we trained our model using datasets fused by multiple sequencing technologies,
excluding the query datasets for testing in each fused dataset. Specifically, in our cross-
platform experiments, we evaluated our model on PBMC datasets sampled using seven
different technologies: Seq-Well, inDrop-seq, CEL-Seq, 10Xv2, Drop-Seq, Smart-Seq2, and
10Xv3. To clarify, when evaluating one of the sequencing datasets, we integrated the
remaining six datasets into a single fusion dataset. This integration was done in such a way

that no genomic information was lost; genes not present in the original data were represented
 with zero expression in the fused dataset. And it resulted in seven distinct fused datasets. Each
 fused dataset was then used as a reference, while the dataset excluded from the fusion served
 as the query dataset. The model was trained on the reference dataset and subsequently used
 to annotate the query dataset. The results regarding to the Figure 2b in manuscript is shown
 in Table \ref{tab:fusion_accuracy}. Meanwhile, we computed the average values of F1-score,
 ARI, and Cohen's Kappa Score across all experiments and the result are shown in Table
 \ref{tab:fusion_all}.”

**Comment 5:**

Mentioned in Line 252, Fig. 4b should correspond to fig. 4a. Fig. 4b lacks corresponding
 content in the main text and its statistical information isn't necessary; it is suggested to be
 deleted.

**Answer 5:**

We thank the reviewer for their attentive reading and constructive comments regarding Fig.
 4b which shows the cell number and gene number of used datasets.

Upon reviewing the manuscript and the figures in question, we acknowledge that Fig. 4b does
 not have a direct reference in the main text, and its inclusion may not add significant value to
 the understanding of the results presented. Following the reviewer's suggestion, Fig. 4b has
 been removed from the manuscript to maintain clarity and focus. We have ensured that all
 relevant information initially intended to be conveyed through Fig. 4b is either adequately
 covered in Fig. 4a or described comprehensively within the text. And the new Fig. 4 is shown
 below. Thank you so much.

Figure. 8: Modification on Figure 4 where the original Fig4b is deleted.

Manuscript

Content in Section “scGraphformer can accurately model the subtle cellular diversity in the mouse brain.”

Caption of Fig. 4:

“Performance of scGraphformer on large-Scale inter-dataset analysis. (a) Heatmap represents

the confusion matrix of classification results for Rosenberg dataset cell types, with a side-by-
side comparison to other methods. The intensity of the colors correlates with the quantity of
cells accurately or erroneously classified. scGraphformer shows a clear advantage in correctly
identifying cell types, especially within smaller subsets, demonstrating its precision and the
breadth of its efficacy compared to other methods. To keep visual clarity and organization,
CellTypist is excluded since it perform the worst. (b) Bar plot with error bars demonstrating
the superior performance of scGraphformer when trained on the Zeisel dataset and tested on
the Rosenberg dataset, in comparison to other methods over five repeated runs. Each bar
signifies the average accuracy for a given method, while error bars denote the standard
deviation, illustrating the consistency and reliability of scGraphformer, as indicated by its
higher mean accuracy and more concentrated error bars. (c) Bar plot illustrating
scGraphformer's classification accuracy for each cell type within the Rosenberg dataset,
alongside a legend indicating the count of cells per type. scGraphformer maintains high
accuracy levels across various cell types, including those that are less represented."

First Paragraph:

"To demonstrate that scGraphformer can effectively discover cell subtypes within major cell
types that appear to be identical. We use two mouse brain scRNA-seq datasets, namely Zeisel
and Rosenberg and each comprising over 100,000 cells (Zeisel: 145954 cells, Rosenberg:
133435 cells). In this study, we utilized Zeisel as a reference dataset and then used the trained
model to annotate the Rosenberg dataset. Shown in Fig. 4b, scGraphformer achieved the
highest mean accuracy of 95.210% with a standard deviation of 0.710 which indicates not
only superior performance in terms of accuracy but also a reasonable consistency across
different runs. The second-best model, scmap-cell, achieved 94.40%, approximately 1%
lower than scGraphformer. This performance gap underscores scGraphformer's effectiveness
in cell type annotation tasks (Details in Supplementary Table E13)."

**Supplementary Appendix E**

"Details of while training on Zeisel and annotate Rosenberg is shown in `\ref{tab:zeisel_compar}`
where we use accuracy score as metric."

**Comment 6:**

From the confusion matrix in Fig. 4d, scmap-cluster performs better in identifying
macrophage cells with a lower proportion in the dataset than scGraphformer. scmap-cell was
compared in Fig. 2 and Fig. 3 but not in Fig. 4d; please provide scmap-cell's results or explain
its exclusion.

**Answer 6:**

Thank you for your observations regarding the performance of scmap-cluster and
scGraphformer in identifying macrophage cells, as illustrated in Fig. 4d. Your point about the
absence of scmap-cell in this figure, despite its comparison in Fig. 2 and 3, is well-noted. In
response to your comment, we have carefully reviewed the rationale behind the configuration
of Fig. 4d and acknowledge the importance of consistency across figures when presenting
comparative analyses.

The exclusion of scmap-cell from Fig. 4d was primarily due to its significant computational
demands, particularly in the context of large-scale datasets. During preliminary trials, scmap-

cell required over an hour to train on the Zeisel data and annotate the Rosenberg dataset,
 significantly longer than other methods, making it impractical for this study focused on
 efficiency. Additionally, for visual clarity and organization in our confusion matrix, we aimed
 to present six methods in a two-row and three-column format. Given its high time cost and
 our layout goals, scmap-cell was excluded to maintain both analytical efficiency and
 presentation clarity.

However, understanding the importance of comprehensive comparative analysis in enabling
 our readers to make informed assessments about method performance, we have revised Fig.
 4d to include the results for scmap-cell. The experimental setting is as the same while we
 evaluate on scmap-cell. Notably, scmap-cell demonstrated superior performance compared
 to scmap-cluster, which is a significant finding we had previously overlooked. We apologize
 for the initial exclusion of this method, which could have provided a more consistent
 evaluation framework. To maintain clarity and organization in our visual presentation, we have
 continued with the original layout consisting of two rows and three columns in our confusion
 matrix. To accommodate the inclusion of scmap-cell while preserving this format, we have
 removed CellTypist—identified as the least effective method—from the updated figure. The
 revised results are clearly depicted in the new Fig. 4, ensuring both comprehensive data
 representation and visual tidiness.

We believe that these revisions will address your concerns and enhance the clarity and rigor
 of our findings.

Figure. 9: Revised Fig 4

Manuscript

Content in Section “scGraphformer can accurately model the subtle cellular diversity in the mouse brain.”

First Paragraph:

“Then, to better compare scGraphformer's performance on such a large and imbalanced
 cross-platform dataset, we also tested the performance of several methods on each cell type.
 Shown in Fig. 4a, we evaluated their identification results and showed their confusion matrix..
 Comparisons show that scGraphformer is better for processing large datasets in cross-
 platform experiments even if the proportion of cell types varies too much. It's particularly

good at identifying major brain cell types like astrocyte cells, neurons, and oligodendrocyte
cells. Compared to other methods, scGraphformer makes fewer mistakes in mixing up similar
cell types. One common challenge for all methods is telling the difference between
endothelial cells and brain pericytes. Even scGraphformer sometimes mixes these up. In
comparison to scmap-cluster and scmap-cell, scGraphformer appears to have reduced off-
diagonal elements, suggesting fewer misclassifications. However, scmap-cell shows
comparable performance for certain cell types, particularly neurons and microglia.
scGraphformer still has a room for improvement in identifying certain cell types.”

**Comment 7:**

It's unclear where scGraphformer is superior to RandomForest and CellTypist in Fig. 5b.
Quantitative assessment will be helpful.

**Answer 7:**

Thank you for your insightful comments requesting a clearer quantitative comparison of
scGraphformer with RandomForest and CellTypist as shown in Figure 5b of our manuscript.
In response, we have conducted a thorough reevaluation of our experiments focusing
specifically on quantitative metrics across each cell type in the Zheng 68K datasets.

Firstly, considering previous comments and considering the complex cell distribution in the
Zheng 68K dataset, which contains both minor and major cell types, we have added F1-score
evaluations for all methods. And we have replacing the RandomForest method with three
new methods which are scType, scBert and TOSICA. The results are presented in the newly
added Figure 11 which is now incorporated into our updated Figure 5 in the paper. To make
our UMAP visualization results clearer and more intuitive, we have introduced a new figure
(Figure 10) that displays the accuracy for each cell type. For clarity and accuracy, we have
highlighted only our scGraphformer model in this visualization. As evidenced by the figure,
scGraphformer demonstrates stability and strong performance across most cell types. Then,
to ensure robustness, we conducted five independent evaluations, with metrics computed on
split test datasets. We used five different random seeds for dataset splitting, ensuring that
each independent experiment used the same test datasets. The results were obtained by
averaging the accuracies for each cell type across different methods. We emphasize this point
because different test sets can influence results, so we directly calculated the average results
to mitigate this potential bias.

These enhancements and the quantitative assessments detailed above clearly demonstrate
scGraphformer's superior performance in comparison to CellTypist, particularly for the Zheng
68K dataset. The additional analyses not only address the concerns raised but also significantly
bolster the claims of scGraphformer's efficacy in handling complex single-cell RNA-
sequencing data. All updated results will be added to our revised supplementary results. We
appreciate your feedback as it has significantly helped in refining our analysis and
strengthening the manuscript.

Figure 10: Average accuracy on each cell type. This detailed bar chart provides a granular view of scGraphformer's accuracy across different cell types. It offers valuable insights into the model's performance for specific cell populations.

Figure 11: F1-score of evaluation on Zheng 68K, this boxplot clearly shows the distribution of F1-scores across different methods, allowing for easy comparison. It effectively demonstrates scGraphformer's performance relative to other methods.

**Manuscript**

**Content in Section "scGraphformer can be applied to large-scale datasets to identify multiple cell types."**

Second Paragraph:

"To benchmark scGraphformer's performance, we compared it with scBalance, and scVI, which epitomize the spectrum of deep learning, traditional machine learning, and canonical cell annotation approaches in handling large-scale datasets. To mitigate the influence of less informative genes, we employed a HVG selection strategy. ~~We experimented with different selection thresholds (3,000, 4,000, and 5,000 genes) to identify the configuration that yields optimal annotation outcomes.~~ We experimented with different selection thresholds (3,000, 4,000, and 5,000 genes) to identify the configuration that yields optimal annotation outcomes. Also, our evaluation needs a lot of computer memory to work. To make it run efficiently, we

also tested the computation burden of different numbers of genes. We wanted to find the
best number that gives good results without using too much memory. After testing, we found
that using 4,000 genes works best.”

Third Paragraph:

“The cell type distribution in Zheng 68K, as shown in Fig.5c, is notably imbalanced. The UMAP
visualization in Fig.5b illustrates the difficulty in distinguishing original cell types by quantity.
Fig.2b and Fig.5d reveal that scGraphformer excels in accuracy over other methods and
performs notably well with minor cell types. In Zheng 68K dataset, as evidenced by the Fig.5d,
scGraphformer demonstrates stability and strong performance across most cell types.”

**Comment 8:**

Line 286 mentions comparing scGraphformer with scBalance, yet scBalance's results aren't
shown in Fig. 5b. Why are the methods compared in Fig 5b and Fig 5d inconsistent and
different from those in Figs 2, 3 and 4

**Answer 8:**

Thank you for your comment highlighting the inconsistency in the method comparisons
across Figures 5b, 5d, and Figures 2, 3, and 4 of our manuscript. We acknowledge that the
differences in the methods compared in these figures may have caused confusion, and we
appreciate your insights which have prompted us to address this issue.

In the initial manuscript, Figure 5b displayed fewer methods to enhance clarity and focus for
that specific dataset. We excluded scBalance to avoid visual clutter. And now we have included
scBalance and fixed some format mistakes in the updated Fig5b. The figure is shown below.

Figure. 12: Modification on Fig 5b.

In contrast, Figure 5d omitted methods like scVI, scmap, and ACTINN from the COVID-19
Atlas analysis due to unique data requirements, though these methods were used in the
Human Neo Cortex Atlas analysis. To improve consistency and clarify our methodological
approach, we re-assessed the application of all methods, including scVI, scmap, and ACTINN,
across both the COVID-19 and Human Neo Cortex Atlases. This adjustment ensures all
relevant methods are consistently compared in our revised figures. As we mentioned at first,
some methods are not designed for large-scale datasets; therefore, we made efforts to adapt

these methods for testing on such datasets with minimal modifications to their core
 algorithms. Meanwhile, due to limitations with our GPU memory (A40, 48GB) and memory of
 HPC, we pre-selected 4,000 highly variable genes (HVGs) in the COVID-19 Atlas to better
 accommodate our hardware capabilities. After these initial processes, we reevaluated the
 models. Below figure shows our final annotation accuracy on Covid Atlas and Human
 Neocortex Atlas. We also compared the time consuming of the best-three models
 (scGraphformer, scBert and CellTypist) and found that scBert needs a lot of time for inference
 in large-scale datasets. The time comparison between the three models is also shown below.

 Figure. 13: Comparison on two atlas

 Figure. 14: Time consuming comparison on three large-scale datasets

We've updated our figures to show the same set of methods across all datasets. This change
 could make our comparisons more consistent and our conclusions stronger. Now, you can
 easily see how each method performs across two atlas. These updates address the concerns
 raised and offer a clearer comparison of the computational methods in our study. And it
 improved presentation helps highlight the strengths and weaknesses of scGraphformer
 compared to other methods like scBalance across various large-scale biological datasets.

Thank you again for your constructive feedback, which has significantly improved the quality
and clarity of our manuscript.

**Manuscript**

**Content in Section “scGraphformer can be applied to large-scale datasets to identify
multiple cell types.”**

Third Paragraph:

“The narrow accuracy margins in this atlas, which is characterized by distinct cellular
composition and expression patterns, suggest a less challenging environment for cell type
annotation and consequently, a subtler display of scGraphformer's strengths. Complementing
these accuracy metrics, Fig. 5g offers valuable insights into the computational efficiency of
scGraphformer compared to other leading methods. The runtime analysis across three
datasets of increasing complexity - Zheng68k, Covid Atlas, and Human Neocortex Atlas -
reveals scGraphformer's superior performance in terms of processing speed. As we progress
to the more complex Covid Atlas and Human Neocortex Atlas, the efficiency gap widens
significantly. scGraphformer maintains consistently low runtimes, even as dataset complexity
increases, whereas both CellTypist and scBERT show substantial increases in processing time.”

**Comment 9:**

scGraphformer's performance is limited on large-scale datasets, with significantly lower
accuracy compared to other datasets, as shown in Figure 2b. This is doubtful about the
performance if compared to large language model-based methods specifically designed for
large-scale datasets.

**Answer 9:**

Thank you for your insightful observation regarding the performance of scGraphformer on
large-scale datasets as depicted in Figure 2b. Your comment highlights an important aspect
of our study: comparing scGraphformer's efficacy against methods specifically tailored for
large datasets.

In fact, our initial motivation for deploying scGraphformer was to address the limited
scalability of traditional graph models, which struggle with extensive datasets like those found
in single-cell genomics. In our manuscript, we specifically chose not to include recent graph-
based models such as scGNN, scGCN, or scGraph in our evaluation because these models are
primarily designed for clustering or imputation predictions, rather than handling large-scale
datasets directly. And they are not specifically designed for annotation task. Instead, we tested
traditional graph models like GAT, GNN, GCN, and variants such as GATJK, GCNJK, and
GPRGNN, along with h2gcn which could treat cell annotation as a semi-supervised node
classification task. None of these models could manage million-level, large-scale single-cell
datasets, nor were they specifically designed for single-cell data, which could render their
evaluation scientifically less meaningful for our purposes. Therefore, our primary goal was to
develop a scalable graph model that could effectively manage large datasets while
maintaining accuracy. While scGraphformer may not significantly outperform all other models,
its framework, which utilizes a graph transformer model, represents a promising approach.
Moving forward, we plan to apply this scalable framework to single-cell multi-omics datasets.

This approach will allow us to construct a more complex heterogenous graph that connects
various biological entities. We believe that the computational costs and potential risks of
losing biological information can be mitigated by our graph transformer-based model,
ultimately enhancing the model's utility and applicability in complex biological analysis. Sorry
for stressing it, I am not arguing with you, I just want to discuss the potential implications of
our model with you.

However, we still valued your comments so much which are so helpful for us. In response to
your feedback, we have incorporated an evaluation of scBert, a large language model-based
method designed for handling large-scale datasets, into our analysis. This inclusion aims to
provide a more comprehensive comparison and better contextualize the capabilities of
scGraphformer within the scope of existing advanced methods. In scBert, we evaluated it on
our two large-scale datasets, Covid Atlas and Human Neo Cortex.

Our revised results, now including scBert, are presented in the updated version of Fig. 5 (also
updated in Fig 2). This addition allows us to directly compare scGraphformer's performance
with a specialized large-scale dataset method, offering a clearer benchmarking against state-
of-the-art technologies in this domain. And in our evaluation on Covid Atlas and Human
Cortex Atlas, we also added scBert as a evaluation method. The previous excluded methods
are also added to evaluate on our updated figure.

We believe that this enhancement addresses your concerns and enriches our manuscript by
demonstrating how scGraphformer stands in relation to specifically designed large language
model-based methods. This comparison not only clarifies the relative performance of
scGraphformer but also underscores our ongoing efforts to refine and adapt our approach
to meet the challenges presented by large-scale datasets.

**Comment 10:**

In Supplementary Fig.E2, the colors for each cell type aren't unique, causing confusion.

**Answer 10:**

Thank you for your feedback regarding the color coding in Supplementary Fig. E2. I
understand your concern about the potential confusion caused using identical colors for
different subtypes within the same major cell type.

In this figure, we used the same color to denote subtypes that belong to the same major cell
type. This was intended to emphasize the relationship between subtypes and their primary
classification group. For instance, B_c01-TCL1A and B_c03-CD27-AIM2 are both colored
similarly because they belong to the major B cell type, indicating their close genetic expression
profiles despite having distinct cellular functions and states. We recognize that this situation
could make it challenging to distinguish between these subtypes briefly, especially when using
classification models. However, this coloring strategy was chosen to highlight the hierarchical
nature of cell type classification in the context of the COVID-19 Atlas, where understanding
the broader lineage relationships can be as crucial as identifying subtle differences among
subtypes. To address the confusion, we add additional contexts under this supplementary

figure E2 which is as the same as the illustration under below figure. We appreciate your
 comment so much as it helps us improve the clarity and usefulness of our visual presentations
 in the manuscript.

 Figure. 15: This figure illustrates the cell type distribution within the COVID-19 Atlas, showcasing
 the diversity and prevalence of each subtype. Here, each bar represents a specific sub-cell type,
 with a total of 64 unique subtypes used as training labels during our model development. The
 colors assigned to each bar indicate the major cell type to which each sub-cell type belongs,
 facilitating an understanding of their broader biological groupings. Each cell concludes its sub
 cell types which more similarity than others. While training, we used sub cell type as training
 labels and there are 64 cell types in total.

 **Comment 11:**

The results of Fig. 5e are confusing. Intuitively, the annotation results of two classes are
 obviously inconsistent with ground truth, but Fig.2d shows that scGraphformer has an
 accuracy of more than 0.93. The model was trained using subtype labels, but how is the
 accuracy calculated? Did the authors calculate the accuracy of 64 sub cell types or the
 accuracy of 12 major cell types? Should the calculation of accuracy be consistent with the
 labels used for training?

**Answer 11:**

Thank you for your insightful observation. We appreciate the opportunity to clarify these
 points.

 Regarding the inconsistency between Fig. 5e and Fig. 2d, we have updated Fig. 2d in our
 revision to address the annotation issue for Baron Human dataset, which was pointed out in
 a previous comment. This update figures in comment 1 resolves the apparent discrepancy
 between the visual results and the reported accuracy. For accuracy calculation, our accuracy

metric is calculated based on the results from the test datasets. However, for the initial
visualization in Fig. 2d (before updated), we used a confusion matrix derived from the
prediction results. We acknowledge that this approach may have led to some inconsistencies.
In our revision, we have standardized our evaluation method to ensure consistency across all
reported metrics by calculating the accuracy or other metric on test dataset. For the next
concern, on subtype or major cell type, our model was trained using the 64 subtypes as labels,
and the accuracy reported in Fig. 5 reflects the performance on these 64 subtypes. We have
emphasized this point in the updated Fig. 5 to avoid any confusion.

The accuracy calculation is consistent with the labels used for training. To be more specific,
before our training, the datasets were randomly split. But this may cause the issue that some
of minor cells are not concluded in the test dataset. And we found this problem during
revision, so we selected StratifiedShuffleSplit from sklearn to make sure the cell type
distribution in test data consistent with dataset. And we think this may help to keep the
consistence. Meanwhile, we added more comprehensive evaluations of our model's
performance, here we introduced additional metrics in our revision, including F1-score,
precision, recall, balanced accuracy etc. These metrics were calculated using standard
implementations from the sklearn package.

For accuracy calculation method, we applied consistently to both baseline models and our
approach, is as follows: "First, it creates a Boolean array called correct by comparing y_true
and y_pred elementwise. This array will contain True for each position where the predicted
label matches the true label, and False where they don't match. Then, it calculates the accuracy
by: a. Summing up all the True values in the correct array using np.sum(correct). This gives
the total number of correct predictions. b. Dividing this sum by the total number of predictions
(which is equal to the length of the correct array). c. Converting the result to a float to ensure
precision in the decimal places. Finally, it returns this calculated accuracy value." We have
shown the code as our attached figure and we add a detailed explanation in our revised
manuscript. We hope these clarifications address your concerns and provide a clearer
understanding of our methodology and results. The consistency between our training labels,
accuracy calculations, and reported results is maintained throughout our study.

```
def eval_acc(y_true, y_pred):  
    correct = y_true == y_pred  
    acc = float(np.sum(correct) / len(correct))  
    return acc
```

Figure. 16: Accuracy evaluation method

Manuscript

Content in Section "Evaluation settings" in Method

First Paragraph

"We discussed our evaluation metrics for cell annotation performance and experiment
settings for scGraphformer and comparison methods. We employed accuracy, F1-score and
Cohen's Kappa Score to evaluate the performance of each method on cell type annotation in
~~intra-experiments and introduced macro-F1 as an additional metric to assess performance~~
~~on minor cell types~~, providing a more comprehensive evaluation of scGraphformer's

capabilities. To conduct a more comprehensive evaluation of our cross-platform experiments,
we introduced Normalized Mutual Information (NMI), Adjusted Rand Index (ARI), Balanced
Accuracy, Precision and recall. These metrics allow us to thoroughly examine the performance
of scGraphformer and other methods across different aspects of cell type annotation tasks.
All calculated using the scikit-learn package: **(Omitted parts using formula for each metrics)**"

Second Paragraph

"We conducted each experiment five times with independent random splits of the datasets,
maintaining the same proportions for input into the model. The results were recorded, and
we calculated the standard error and mean accuracy to ensure the reliability and
reproducibility of our findings. While scGraphformer can operate without a predefined graph,
we experimented with incorporating a KNN-graph for each dataset during annotation.
Contrary to our initial expectations, the addition of the KNN-graph did not significantly
improve performance. This result suggests that scGraphformer's base architecture already
captures essential cell relationships effectively (Details shown in Supplementary Fig. G5). The
limited impact of the KNN-graph hints at the potential for more biologically relevant graph
structures. We hypothesize that graphs based on known cellular interactions, developmental
trajectories, or functional relationships might provide more meaningful improvements."

**Comment 12:**

In the Methods of "The structure of scGraphformer", authors said "When there is a high-
quality cell graph, we can embed a GCN module in the scGraphformer to incorporate the
structure information." From my understanding, If no prior cell graph is known, cell
embedding z is updated according to equation (6) via Q,K,V transformation. However, if a
high-quality cell graph exists, cell embedding z is updated according to equation (8), which
has nothing with the attention mechanism. Is my understanding correct? It's better to make
it more clear in the methods. For my curiosity, how to evaluate whether a cell graph is of high
quality?

**Answer 12:**

Thank you for your insightful comments and questions. They have prompted us to clarify
certain aspects of our method and its underlying assumptions more clearly in our manuscript.

In our design, the scGraphformer framework, which is based on a lightweight transformer
structure, can capture global information between nodes in the graph where the Graph
Convolutional Network (GCN) module integrates cell topology structures alongside gene
expression features. This topology acts as a relational inductive bias within our transformer
model, enhancing its ability to model complex biological relationships. Our initial goal was to
move beyond traditional Graph Attention Networks used in single-cell analysis, which
primarily focus on local relationships between cell nodes, by incorporating a graph
transformer structure. However, scRNA-seq analysis presents unique challenges that must be
considered in model design due to several biological realities: 1. The actual biological
implications of cell topology can differ significantly from the KNN-constructed topology. 2.
The scalability requirements of recent single-cell atlases impose significant computational
costs. To address these issues, we have refined our transformer's computation of Query, Key,
and Value to more aptly represent the cell nodes within the graph. Additionally, to manage

the computational demands of graph learning, we adopted a lightweight transformer
approach. This involves replacing the traditional SoftMax function with a Taylor expansion of
the exponential SoftMax function, which reduces the time complexity of the attention
mechanism from $O(N^2)$ to $O(N)$. By integrating cell topology, our model incorporates the
graph structure directly into its calculations. The GCN module introduces the 'Value'
component in the transformer. Subsequently, the results from the GCN are propagated to
the transformer, where they inform the updating of next-level embeddings.

You are correct in noting that our method updates the cell embedding z differently based on
the availability and quality of a pre-existing cell graph. We want to clarify: 1. Without a high-
quality cell graph, the cell embedding z is updated according to Equation (6), where the
transformation relies on the computation of Query (Q), Key (K), and Value (V) within the
transformer's attention mechanism; 2. With a high-quality cell graph, the cell embedding z is
updated according to Equation (8), where the GCN module introduces structured information
into the Value component, and this structured information is then integrated within the
transformer framework. This process utilizes the attention mechanism, and I apologize for any
previous ambiguity in this explanation.

Regarding your question about how we evaluate whether a cell graph is of high quality, we
initially considered using the heterophily of the graph as a quality metric. Through further
examination, we realized that our primary objective was to capture and utilize biological
topology information, specifically focusing on the relationship between gene expression
patterns and biological graphs, such as cell topology in spatiotemporal transcriptomics. This
realization led us to adopt the KNN-constructed graph as a standard due to its relevance and
practicality, even though it may not perfectly represent all biological interactions. We
acknowledge that our task was initially limited to cell type annotation and that no specific
scRNA-seq datasets were available to evaluate the relationships we were interested in
exploring. This limitation led us to rely on assumptions and a simplified model. We appreciate
your curiosity and agree that exploring how to accurately assess the quality of cell graphs in
this context remains an open and intriguing area for future research.

Thank you again for your constructive feedback, which has significantly helped improve our
manuscript and refine our approach. We hope that our revisions and responses address your
concerns adequately.

**Comment 13:**

Overall, cell type annotation for scRNA-seq is a well-studied topic, the significance is
somehow limited, but the method is of some novelty.

**Answer 13:**

Thank you for your thoughtful comment regarding the scope and significance of our study
on cell type annotation for single-cell RNA sequencing (scRNA-seq). We appreciate your
recognition of the novelty in our method.

While it is true that many approaches exist in this well-explored field, we believe that the

continuous development of new methodologies can still yield significant improvements in
both accuracy and efficiency, which are critical as the scale and complexity of single-cell
datasets grows. Besides, existing methods often struggle with specific challenges, such as
reliance of pre-defined graph and scalability.

The proposed framework addresses these limitations by leveraging a graph transformer-
based model, which allows for a more nuanced representation of cell types and their
relationships. To the best of our knowledge, *scGraphformer is the first method that can learn*
*an all-encompassing cell-cell relational network directly from scRNA-seq data.*

We believe that these innovations not only enhance the accuracy and utility of cell type
annotation but also lay a solid foundation for future research. Our approach could be adapted
to tackle similar challenges in other areas of single-cell analysis or across different omics data
types.

We hope that our manuscript sufficiently highlights these contributions, underscoring the
relevance and potential impact of our work amid the existing body of research. Thank you
once again for your insightful feedback, which has given us the opportunity to clarify the
significance and novelty of our study.

**Reviewer #2:**

**Comment:**

In this manuscript, Fan et al. propose a method for cell type identification based on single-
cell RNA sequencing data, namely scGraphformer. This is a Transformer-based GNN model
that overcomes the classical GNN models' reliance on kNN by directly integrating the cell
interaction network from the gene-cell matrix. Through an iterative optimization process,
scGraphformer constructs a graph structure that captures all cell interaction relationships. In
the experiments, the authors demonstrate scGraphformer's ability to accurately annotate cell
types both within and between datasets, proving its superiority in accuracy over existing
methods. Additionally, the authors show its scalability on large datasets and its significance in
revealing biological insights. The method is novel and interesting, and while there have been
many GNN-based scRNA-seq data analysis models, the main contribution of this manuscript
is the introduction of the Transformer mechanism and the abandonment of the kNN
predefined graph to capture complex intercellular dependencies.

However, there are still many issues that require further explanation or improvement and the
authors must address our concerns regarding the data and figures in the manuscript (Major
Comment #4). Moreover, the manuscript contains an extremely large number of basic errors,
which seem not to have been carefully checked and corrected. When revising the manuscript,
the authors must meticulously check the details in the article and ensure contextual
consistency.

**Answer:**

Thank you for your constructive feedback and the positive remarks on the novel aspects of
our scGraphformer model. We appreciate your recognition of the innovative integration of
the Transformer mechanism and the development of a model that does not rely on a
predefined kNN graph, which allows for capturing complex intercellular dependencies more
effectively. Our motivation is to leverage the graph transformer's ability on capturing global
interaction between cell nodes which overcomes the limitations of GAT-based or GCN-based
method. Previous methods for cell type annotation using KNN graphs are inherently
constrained by the limitations of predefined neighborhood relationships, which often
overlook complex and global cellular interactions. Our method, scGraphformer, is designed
to address this critical gap by incorporating a graph transformer architecture. This innovative
approach allows for a more dynamic and comprehensive analysis of cell interactions, enabling
a deeper understanding of cellular functions and relationships. We are grateful for your
recognition of this significant advancement, and we sincerely appreciate your
acknowledgment of our contribution to improving cell annotation methodologies.

We are immensely grateful for your insightful feedback, which has highlighted several critical
issues in our manuscript. Through our revision this time, we have found a lot of errors and we
apology for our careless and recklessness. And we acknowledge our mistakes and basic errors
in our manuscript, and we hope our following manuscript could fix those errors and problems.
We deeply appreciate your guidance and the time you have taken to help us improve our
work. We are hopeful that the revisions made will address your concerns satisfactorily.

In response to your feedback and guidance, we have carefully revised our manuscript,
addressed each error and improving the overall presentation and accuracy. Regarding
Comment #4, we have detailed our responses and the modifications made below, which are
also clearly marked in the revised manuscript. We apologize for the carelessness that led to
many of these errors, and we appreciate your patience and the thoughtful insights that have
significantly helped in refining our work. After our thorough revision, we believe that the
corrections and improvements made not only resolve the issues you pointed out but also
reinforce the credibility and relevance of our model. We trust that these changes will reassure
you of the value and validity of our approach.

We sincerely hope that these revisions meet your expectations and significantly improve the
manuscript. We are committed to ensuring the highest quality of our work and are grateful
for the opportunity to refine our submission further. Thank you once again for your valuable
feedback.

**Major Comments:**

**Comment 1:**

The authors frequently use “accuracy” as the evaluation metric for different methods. However,
as shown in the manuscript (Fig. 2d, Fig. 5a), there is a significant disparity in the number of
cells across different cell types. In such cases, using “accuracy” as an evaluation metric is
inadequate to reasonably assess the model's ability to identify rare cell types. The authors
need to employ additional evaluation metrics such as Kappa, precision, recall, and F1 score
for benchmarking.

Furthermore, the authors claim to use the macro F1 score as an additional evaluation metric
(line 494), but the corresponding results are not presented in the manuscript. Additionally,
the article claims to use "mean accuracy" to evaluate the model's performance on each cell
type (Fig. 2c); however, it seems impossible to calculate "mean accuracy" for an individual cell
type. The authors need to provide more information to clarify the evaluation pipeline used in
the article

**Answer 1:**

We sincerely appreciate your thorough review and insightful comments regarding our
evaluation metrics. Your feedback has been so valuable in improving the clarity and
robustness of our analysis.

In response to your suggestions, we have expanded our evaluation framework to include a
comprehensive set of metrics beyond accuracy. We now incorporate: Adjusted Rand Index
(ARI), Balanced Accuracy, Cohen's Kappa Score, F1-Score, Normalized Mutual Information
(NMI), Precision, and Recall. These metrics are implemented using the standard sklearn
package, ensuring consistency and reliability. We've applied these metrics uniformly across
scGraphformer and all comparative methods. A detailed explanation of each metric's usage
and significance has been added to the revised manuscript (Section 4.5: Evaluation Setting).

We acknowledge the oversight in our previous version regarding the F1-Score. This metric
has now been fully incorporated into our analysis, providing a robust measure of
scGraphformer's performance on imbalanced datasets. The complete results are now
included in supplementary, with additional details available in the supplementary material.
Below shows our results using different metrics (here we list F1-Score and its table results).
TOSICA and scType models are not included in our plot since their results are bad in our
evaluation.

Regarding the 'mean accuracy' mentioned in Fig. 2c, we appreciate the opportunity to clarify.
This term refers to the average accuracy for each cell type across five independent
experimental repeats, not the Mean Accuracy calculation as it might have been misinterpreted.
We have updated our manuscript to clearly articulate this methodology, avoiding any
potential confusion. We have thoroughly revised our 'Evaluation settings' section to reflect
these changes and provide a more comprehensive overview of our analytical approach.

Once again, we thank you for your critical feedback. It has significantly enhanced the quality
and clarity of our work, allowing us to better demonstrate the significance and novelty of our

method	scGraphformer	ACTINN	CellTypist	RandomForest	scBalance	Bert(Finetune)	scBert(Zero-Shot)	scVI	scmap-cell	scmap-cluster
AMB	80.7 ± 2.87	33.72 ± 2.67	92.29 ± 0.22	86.93 ± 0.24	86.88 ± 1.42	nan ± nan	nan ± nan	34.89 ± 1.23	88.27 ± 0.0	92.06 ± 0.0
Adam	97.59 ± 0.58	93.56 ± 1.68	96.54 ± 0.35	95.06 ± 0.83	96.11 ± 0.71	nan ± nan	nan ± nan	96.75 ± 0.1	80.99 ± 0.0	91.37 ± 0.0
Bach	98.92 ± 0.16	90.34 ± 1.77	97.89 ± 0.05	98.21 ± 0.05	99.01 ± 0.09	nan ± nan	nan ± nan	95.03 ± 2.5	97.7 ± 0.0	98.0 ± 0.0
Baron Human	98.63 ± 0.25	91.59 ± 3.76	97.19 ± 0.1	96.93 ± 0.12	98.54 ± 0.07	90.26 ± 0.97	20.12 ± 0.0	94.16 ± 2.0	97.99 ± 0.0	97.55 ± 0.0
Baron Mouse	99.05 ± 0.71	96.35 ± 0.47	95.77 ± 0.67	96.67 ± 0.49	97.38 ± 0.09	nan ± nan	nan ± nan	92.36 ± 2.44	98.95 ± 0.0	97.66 ± 0.0
Deng	98.19 ± 1.8	95.81 ± 1.95	90.1 ± 1.24	96.22 ± 1.23	90.64 ± 5.19	nan ± nan	nan ± nan	81.35 ± 9.08	98.15 ± 0.0	85.01 ± 0.0
Klein	99.34 ± 0.46	95.21 ± 4.7	99.41 ± 0.18	98.11 ± 0.14	99.11 ± 0.34	nan ± nan	nan ± nan	98.88 ± 0.12	90.83 ± 0.0	94.25 ± 0.0
Muraro	97.7 ± 0.4	98.03 ± 0.19	97.42 ± 0.33	98.21 ± 0.28	98.95 ± 0.15	nan ± nan	nan ± nan	97.08 ± 0.15	96.94 ± 0.0	97.43 ± 0.0
Segerstolpe	98.76 ± 0.43	93.98 ± 1.01	95.06 ± 0.47	96.02 ± 0.26	97.05 ± 0.21	85.4 ± 1.12	1.11 ± 0.0	92.48 ± 0.12	95.72 ± 0.0	97.79 ± 0.0
TM	94.42 ± 0.26	62.3 ± 2.95	96.01 ± 0.03	95.63 ± 0.05	94.01 ± 0.16	nan ± nan	nan ± nan	68.75 ± 3.09	96.24 ± 0.0	90.35 ± 0.0
Tosches_turtle	96.3 ± 0.21	73.42 ± 3.08	94.38 ± 0.23	94.45 ± 0.16	96.5 ± 0.21	96.75 ± 0.02	17.56 ± 0.0	91.28 ± 0.11	94.88 ± 0.0	95.55 ± 0.0
Xin	100.0 ± 0.0	89.23 ± 5.28	94.05 ± 0.58	96.56 ± 0.83	99.78 ± 0.27	86.15 ± 0.0	32.68 ± 0.0	99.08 ± 0.37	99.65 ± 0.0	100.0 ± 0.0
Young	97.5 ± 0.6	82.18 ± 4.98	95.96 ± 0.27	92.82 ± 0.12	94.89 ± 0.57	97.38 ± 0.04	15.34 ± 0.0	85.35 ± 7.82	94.7 ± 0.0	90.05 ± 0.0
Zheng 68K	74.11 ± 0.33	24.52 ± 7.75	72.38 ± 0.09	68.43 ± 0.14	65.65 ± 2.32	75.89 ± 0.83	14.12 ± 0.0	69.5 ± 1.07	54.54 ± 0.0	62.5 ± 0.0
campLiver	99.23 ± 0.84	96.78 ± 0.56	98.72 ± 0.0	97.12 ± 0.55	98.35 ± 0.21	71.51 ± 8.75	1.29 ± 0.0	93.12 ± 6.33	92.96 ± 0.0	96.7 ± 0.0
campbell	89.37 ± 0.7	45.92 ± 1.88	77.46 ± 0.49	81.99 ± 0.37	88.67 ± 0.25	nan ± nan	nan ± nan	54.14 ± 3.89	82.93 ± 0.0	89.93 ± 0.0
darmanis	92.35 ± 3.25	90.15 ± 1.28	80.99 ± 0.55	82.15 ± 2.3	76.33 ± 6.47	11.99 ± 0.0	2.0 ± 0.0	72.16 ± 3.01	88.9 ± 0.0	87.85 ± 0.0
lake	92.49 ± 0.74	62.6 ± 5.14	86.97 ± 1.26	91.83 ± 0.43	90.07 ± 0.76	91.9 ± 0.44	18.06 ± 0.0	73.82 ± 4.67	81.14 ± 0.0	90.98 ± 0.0
usoskin	99.03 ± 0.67	96.49 ± 1.03	98.04 ± 0.4	94.65 ± 0.38	96.95 ± 0.56	nan ± nan	nan ± nan	96.09 ± 1.0	90.8 ± 0.0	94.91 ± 0.0
zillionis	97.75 ± 0.1	82.51 ± 2.35	96.57 ± 0.06	95.5 ± 0.19	97.32 ± 0.32	98.32 ± 0.06	20.43 ± 0.0	93.72 ± 0.53	94.81 ± 0.0	95.43 ± 0.0

Table 2: F1-score is added in to our supplementary material.

Supplementary Appendix C

First Paragraph

“All of comparison results are shown in Table C2. Each method is running five times and the value represents its mean accuracy and standard deviation. In Table C5, it shows all five experiments' results on scGraphformer. Table C6 shows comparison of each method on baron human datasets, and each column represents cell type. Besides accuracy, we have also compared the F1-score and Cohen's Kappa score in intra-datasets evaluation which are shown in Table C3 and Table C4.”

Comment 2:

With the advancement of sequencing technologies, the scale of the dataset obtained from a single sequencing run has increased exponentially. It is encouraging to see that scGraphformer can handle such large-scale datasets. Nevertheless, the authors still need to provide a more comprehensive comparison of run-time and memory usage between scGraphformer and other baseline methods. In Fig. 5d, the authors only compared two baseline methods and did not explain why other methods were not run. Moreover, scGraphformer does not demonstrate a significant advantage on these two datasets. Therefore, if the runtime of scGraphformer is significantly higher than other comparison methods, it would be difficult to convince readers to choose scGraphformer for large-scale datasets.

Answer 2:

Thank you for sharing this comment from your paper revision. It's valuable feedback that highlights the need for a more comprehensive comparison of computational performance.

Large-scale models are naturally designed for annotating large-scale datasets; therefore, we document the running-time on Covid Atlas and Human Cortex Atlas. Beside Covid Atlas and Human Cortex Atlas, we also evaluate the time on Zheng 68K since it's one of classical datasets. The running time comparisons were recorded in the same device where we evaluated the Zheng 68K dataset on RTX 4090 (24GB), and two atlas were evaluated on NVIDIA A40 (48GB). We picked the top three models that performed best to compare the

1020 running-time efficiency. And below figure shows our evaluations and comparisons. We
understand that scGraphformer may not show significant advantages in runtime for the
datasets in Fig. 5d. However, its true strength lies in its ability to handle much larger datasets
that are becoming increasingly common in single-cell research. We believe this capability
justifies potential increases in computational time for certain scenarios.

Figure. 17: Comparison on two atlas

In terms of memory usage, there is a significant difference among various methods. Classic methods, such as CellTypist, do not require much memory. In contrast, LLM-based methods typically demand a larger memory capacity. Our proposed method, scGraphformer, has moderate memory consumption; furthermore, when used without an input graph, its memory usage can be reduced even further. These characteristics highlight its capability to handle large datasets effectively.

For another problem that we only compared two baseline methods in Figure 5d. We
acknowledged that and apologized that we didn't include reasons in our manuscript. In our
original manuscript, we omitted methods like scVI, scmap, and ACTINN from the COVID-19
Atlas analysis since they cannot handle such a large data. However, we acknowledge our
inconsistency and may confuse our readers. Following your suggestion, we re-assessed the
application of all methods to improve our consistency and clarify our methodological
approach, including scBert, scVI, scmap-cluster, and CellTypist, across both the COVID-19
and Human Neo Cortex Atlases. As initially noted, several methods were not originally
designed for large-scale datasets. We made efforts to adapt these methods for testing on
such datasets, implementing minimal modifications to their core algorithms to maintain their
integrity. However, two methods were ultimately excluded from our analysis: ACTINN, due to
its inherently memory-intensive framework design, and scmap-cell, which we had to
discontinue after it failed to produce predictions despite running for 32 hours. And this
adjustment ensures most of relevant methods are consistently compared in our revised
figures. And the results are shown below and we had added it to our new Fig. 5.

Figure. 18: Running time comparison on three large-scale datasets

Thank you so much for your advice. Your feedback has highlighted valuable areas for potential improvement, for which we are grateful. In future iterations, we plan to further optimize scGraphformer's runtime performance for multi-omics analysis. Notably, in the multi-omics field, many graph-based methods such as SIMBA primarily rely on CPU computing without GPU acceleration. Our next step is to apply the graph transformer model to replace these conventional packages. We anticipate that scGraphformer's value in graph analysis will significantly increase, particularly when dealing with large-scale, complex multi-omics datasets. Thank you so much!

Content in Section “scGraphformer can be applied to large-scale datasets to identify multiple cell types.”

Third Paragraph:

“The narrow accuracy margins in this atlas, which is characterized by distinct cellular composition and expression patterns, suggest a less challenging environment for cell type annotation and consequently, a subtler display of scGraphformer's strengths. Complementing these accuracy metrics, Fig. 5g offers valuable insights into the computational efficiency of scGraphformer compared to other leading methods. The runtime analysis across three datasets of increasing complexity - Zheng68k, Covid Atlas, and Human Neocortex Atlas - reveals scGraphformer's superior performance in terms of processing speed. As we progress to the more complex Covid Atlas and Human Neocortex Atlas, the efficiency gap widens significantly. scGraphformer maintains consistently low runtimes, even as dataset complexity increases, whereas both CellTypist and scBERT show substantial increases in processing time.

Comment 3:

For the inter-dataset experiments, the authors paired two datasets, using one as the reference dataset to train the model and the other as the query dataset to test the model's performance. However, the authors suggest that using 60% of the reference dataset for training better reflects real-world scenarios. Since this dataset is fully annotated, why wouldn't using 100% of it be closer to realworld conditions? Can the authors provide more explanation or references for this? Moreover, scGraphformer does not show a significant advantage when using 60% of

1082 the reference dataset for training. Therefore, the authors need to provide the performance of
1083 the baseline methods when using 100% of the reference dataset for training to demonstrate
the advantages of scGraphformer.

**Answer 3:**

Thank you for your insightful comments. We appreciate the opportunity to clarify and provide
additional context.

Firstly, we must apologize for an error in our original manuscript. We actually used 80% of the
reference dataset for training, not 60% as initially stated. This has been corrected in our revised
manuscript. We sincerely regret any confusion this may have caused. Regarding the use of
80% rather than 100% of the reference dataset, our approach aims to mirror real-world
scenarios where researchers often set aside a portion of their data for validation. This practice
helps ensure model generalizability and prevents overfitting. In our evaluation, we split the
reference dataset into 80% for training and 20% for validation. While using 100% of the data
for training is possible, it may not always reflect best practices in model development and
validation. Meanwhile, scGraphformer split the reference dataset into training data and valid
data, the proportion is 80% and 20%, respectively. Though some comparison methods may
not require a separate validation set, to maintain consistency across our evaluations and to
provide a fair comparison, we applied the same 80/20 split to all methods. This could allow
for a more standardized assessment of performance across different techniques.

Your suggestion about using 100% of the reference dataset is insightful and has prompted us
to reconsider our approach especially the real-scenario. While we maintain that the 80/20
split is valuable for model validation, your comment, combined with feedback from another
reviewer, has inspired a new direction in our evaluation strategy. We've decided to implement
a more comprehensive approach by incorporating all other reference datasets. Specifically,
when evaluating any single sequencing dataset, we now integrate the remaining six datasets
into a unified fusion dataset. We believe this approach more closely aligns with real-world
scenarios, where biologists may not always distinctly separate datasets by sequencing
technology. Instead, they often use mixed data to annotate newly sequenced datasets.
Consequently, we've designed a new evaluation framework where these fused datasets serve
as the reference, and we use this model to annotate other methods. This new evaluation may
not only addresses your concern but also enhances the practical relevance of our study. And
our result is shown below. As you can see from the figure, the annotation performance is
significantly improved.

Figure. 19: Comparison on fusion data and single data

Concerning scGraphformer's performance, we acknowledge that it doesn't consistently outperform other methods by a large margin. However, we believe its strength lies in its well-rounded performance across multiple metrics, as shown in the attached figure. While scGraphformer may not dominate in every category, it consistently performs at or near the top across all evaluated metrics. This balanced performance demonstrates its versatility and robustness across different aspects of cell type annotation tasks. To provide a more comprehensive view, we've included additional evaluation metrics in our analysis (as shown in the figure). These metrics offer a broader perspective on each method's strengths and weaknesses. We believe this holistic approach to evaluation is crucial for understanding the true capabilities of different annotation methods in varied scenarios.

We appreciate your critical feedback, as it helps us improve the clarity and rigor of our research. We remain committed to providing a fair and comprehensive comparison of cell type annotation methods, and we welcome further discussion on how to best represent real-world scenarios in our evaluations. And the revised context is in added into Appendix D

Figure. 20: updated figure in Fig.3d which is comprehensive comparison of performance metrics across all cross-platform experiments for different methods.

Manuscript

Content in Section “scGraphformer is robust in inter-dataset evaluation”

Caption of Fig. 3:

“(b) Comparative performance of scGraphformer trained on single-platform data versus fusion data across various cross-platform scenarios. The x-axis represents different query datasets, while the y-axis shows the F1-Score, ranging from 85 to 100. Box plots depict the statistical distribution of results for each dataset and data type. Green boxes represent models trained on fusion data, which integrates data from all sequencing technologies except the one being queried. Orange boxes represent models trained on single-platform data. (d) comprehensive comparison of performance metrics across all cross-platform experiments for different methods. Eight metrics are displayed: Accuracy, ARI, Balanced Accuracy, Cohen’s Kappa Score, F1-score, NMI, Precision, and Recall. Each subplot uses box plots to show how the seven methods (scGraphformer in blue, others in gray) perform across all experiments. scGraphformer consistently outperforms other methods in most metrics, especially in Accuracy, ARI, F1-score, and NMI, showing both higher median values and less variation.”

Third Paragraph

Supplementary Appendix D.

First Paragraph

“In this table, we perform our model in this experiment for two settings (set the training
proportion of reference data for 0.8 and 1.0, respectively representing realistic and ideal
situations while performing cross-platform experiments). The results in Table D8 shows that
ideal performance is better that the realistic one. But we finally choose to set training
proportion of reference dataset to 0.8. Regarding the use of 80% rather than 100% of the
reference dataset, we expect to mirror real-world scenarios where researchers often set aside
a portion of their data for validation. This practice helps ensure model generalizability and
prevents overfitting. In our evaluation, we split the reference dataset into 80% for training and
20% for validation. While using 100% of the data for training is possible, it may not always
reflect best practices in model development and validation. We set the same training
proportion on all comparing methods while evaluating them on cross-platforms experiments.
Therefore, we choose the 0.8-proportion as our results on the paper since the demands of
realistic situation. And the results correspond to the UMAP visualization in Figure 2e in
manuscript.”

Second Paragraph

“Meanwhile, we trained our model using datasets fused by multiple sequencing technologies,
excluding the query datasets for testing in each fused dataset. Specifically, in our cross-
platform experiments, we evaluated our model on PBMC datasets sampled using seven
different technologies: Seq-Well, inDrop-seq, CEL-Seq, 10Xv2, Drop-Seq, Smart-Seq2, and
10Xv3. To clarify, when evaluating one of the sequencing datasets, we integrated the
remaining six datasets into a single fusion dataset. This integration was done in such a way
that no genomic information was lost; genes do not present in the original data were
represented with zero expression in the fused dataset. And it resulted in seven distinct fused
datasets. Each fused dataset was then used as a reference, while the dataset excluded from
the fusion served as the query dataset. The model was trained on the reference dataset and
subsequently used to annotate the query dataset. The results regarding to the Figure 2b in
manuscript is shown in Table D11 Meanwhile, we computed the average values of F1-score,
ARI, and Cohen's Kappa Score across all experiments and the result are shown in Table D12.”

**Content in Methods “Evaluation settings”**

Fourth Paragraph

“For cross-platform (inter-dataset) experiments, we used the reference dataset for training
and the query dataset solely for testing. While training, the reference dataset is split into
training data and valid data (0.8/0.2). We retained only the common genes between datasets
to ensure a consistent input feature space. In fusion data experiment, for each evaluation of
a single sequencing dataset, we integrated the remaining six datasets into a unified fusion
dataset. Then we train the model in fusion dataset and tested on the excluded sequencing
data.”

**Comment 4:**

If I understand correctly, Fig. 3c corresponds to the data in Table D5. The authors claim in the
manuscript that scGraphformer's mean accuracy is 96.08% (line 219). However, calculating the
mean accuracy using the data from Table D5 yields 95.915%. Similarly, the median obtained
from Table D5 for the scGraphformer method is 96.625%, while for the CellTypist method, it
is 96.696%. However, in the boxplot in Fig. 3c, the median for scGraphformer is significantly

higher than that for CellTypist. We redrew the boxplot in Fig. 3c using the data from Table
D5, and it seems to differ from what is presented in the manuscript. Could the authors provide
more explanation or supplementary material for this data?

**Answer 4:**

We sincerely appreciate your thorough review and insightful comments. Your observations
have led us to conduct a comprehensive re-evaluation of our results, for which we are grateful.

Regarding the discrepancies you noted, we apologize for any confusion this may have caused.
Upon careful re-examination, we discovered that the initial inconsistencies were due to
manual calculation errors on our part. To address this issue and ensure the highest standard
of accuracy, we have taken the following steps: 1. We have re-evaluated all our experiments
using automated programmatic calculations to eliminate human error. 2. For consistency and
fairness, we retrained our model using uniform parameters across all experiments, without
specific tuning for individual cases to obtain the best results.

As a result of this rigorous re-evaluation, we have updated our results, plots, and tables
throughout the manuscript. The revised figure and table, which we've included below, reflect
these changes:

Figure. 21: accuracy distribution across all cross-platforms experiments

As evident from the updated data, the mean accuracy of scGraphformer across all
experiments is now 94.01%. The median accuracy is 94.92%, which indeed surpasses the
second-best method. Furthermore, to provide a more comprehensive evaluation, we have
calculated additional metrics including F1-score, ARI, NMI, Cohen's Kappa score, Precision,
Recall, and Balanced Accuracy. In our metrics Across these metrics, scGraphformer
demonstrates superior performance in both median and mean values. These extended results
have been incorporated into our updated supplementary material for a more thorough
analysis. Below comparison table shows three metrics we picked including their median and

mean values.

method	Accuracy			F1-Score			NMI			Cohen's Kappa Score		
	median	mean	std	median	mean	std	median	mean	std	median	mean	std
scGraphformer	94.92	94.01	2.31	94.29	92.83	3.48	86.8	86.71	2.24	93.05	91.77	2.98
scBalance	94.13	93.28	2.19	93.96	92.43	3.42	85.84	85.75	1.97	92.37	91.25	2.84
scVI	92.69	92.4	2.52	92.94	91.35	3.41	86.02	85.86	2.62	90.38	90.04	3.23
CellTypist	93.42	92.26	2.26	90.6	90.31	3.61	83.47	83.03	1.9	91.29	89.88	2.92
scmap-cell	89.61	89.41	1.73	90.07	89	3.01	78.04	77.86	1.58	86.63	86.28	2.23
scmap-cluster	82.92	83.28	1.81	83.78	83.57	2.44	69.64	69.76	1.57	78.27	78.56	2.17
ACTINN	80.69	80.61	2.64	76.69	76.74	3.74	69.21	69.64	1.93	74.46	73.95	3.48

Table 3: Summarizing values comparison across all 42 cross-experiments.

We have revised our manuscript accordingly, ensuring that all claims and figures accurately reflect these re-evaluated experimental results. We believe these changes not only address the concerns raised but also strengthen the overall validity and reliability of our findings. We sincerely thank you for your meticulous review, which has significantly contributed to improving the quality and accuracy of our work. If you have any further questions or require additional clarification, please don't hesitate to ask. We are committed to maintaining the highest standards of scientific rigor and transparency in our research.

Manuscript

Content in Section "scGraphformer is robust in inter-dataset evaluation"

Caption of Fig. 3:

We have write it in reply to Comment 4

Second Paragraph

"Preliminary results, illustrated in Fig. 2a and Fig. 2b, demonstrate scGraphformer's proficiency in maintaining annotation accuracy across datasets when ~~respectively~~ trained on 10Xv2 and ~~10Xv3~~ and tested on other platforms; and when trained on 10Xv2 and tested on Seq-Well, scGraphformer achieved a mean accuracy of 95.46% ~~96.08%~~, which is approximately 2% ~~3.54%~~ higher than the average of ~~other~~ the second best methods (CellTypist: 93.66% , details shown in Supplementary Table D57). During training, genes that were not expressed in both training and testing data were excluded from the feature space, thereby enhancing the method's resilience to batch-induced variability. The results indicate that scGraphformer's ability to overcome batch effect. Furthermore, to compare its performance with other methods under batch effect, we evaluate the annotation performance of all pairwise train-test combinations between 7 protocols on the left methods. During training, genes that were not expressed in both training and testing data were excluded from the feature space, thereby enhancing the method's resilience to batch-induced variability. The results indicate that scGraphformer's ability to overcome batch effect. Furthermore, to compare its performance with other methods under batch effect, we evaluate the annotation performance of all pairwise train-test combinations between 7 protocols on the left methods. All 49 experiment results are summarized in Fig. 2c and the ~~accuracy score~~ Adjusted Rand Index (ARI) score is used as the evaluation metric. And we set training proportion of 80% in reference datasets for realistic situations (details in Supplementary Appendix D and Table D8). Besides ARI-score, we also used accuracy score, Normalized Mutual Information (NMI), F1-score to compute the annotation performance (Supplementary Fig D2, Fig D3 and Table D7). Under each metric, scGraphformer achieves better in overall performance which has better median and mean value across all experiments. (Supplementary Table D10 and Fig. 2d). ~~The percentage improvement of scGraphformer over other methods ranges from a modest 0.08% to as high as 5.18%, depending on the specific reference-query pair (Supplementary Table D6). This~~

variance underscores the method's adaptability to diverse dataset characteristics and
sequencing platforms. The results show that scGraphformer achieved a good performance
across all experiments and there is not much difference from other state-of-art methods.
Moreover, scGraphformer's performance at rare cell type identification was also
demonstrated through cross-platform experiments, where we employed Uniform Manifold
Approximation and Projection (UMAP) for visualization purposes while training on Smart-seq
and testing on 10Xv3 (Fig. 2e and Supplementary Table D97), and it visualizes the clustering
result of scGraphformer with the original true cell types and predicted cell types. This aspect
of the analysis highlighted scGraphformer's superior performance in accurately annotating
minor cell populations, a critical capability given the frequent occurrence of imbalanced
datasets in scRNA-seq studies.”

**Comment 5:**

In Fig. 3d, the UMAP visualization for scGraphformer seems different from the UMAPs of other
baseline methods. Did the authors apply any special processing to the UMAP for the
scGraphformer method, or is this a plotting mistake?

**Answer 5:**

Thank you for bringing this to our attention and for providing the updated figure. We
appreciate your diligence in identifying and correcting this inconsistency. You are correct, and
we apologize for the confusion caused by the original figure. Upon review, we realized there
was an inconsistency in our visualization methods across different approaches. To address
this, we have standardized our plotting methodology by using Seurat in R for all visualizations,
including scGraphformer and the baseline methods.

In the updated figure, we have maintained consistency across all methods. The UMAP
visualizations now show the results of annotating the 10Xv3-seq dataset using the Smart-
seq2 dataset as a reference. The predicted labels from each method were used to map cell
type labels in the UMAP visualization. It's important to note that none of the methods,
including scGraphformer, were able to identify Natural Killer cells and Dendritic cells. This is
because these cell types are not present in the reference Smart-Seq2 dataset, and thus could
not be accurately predicted in the target dataset.

We appreciate your careful review of our work, as it has helped us improve the clarity and
consistency of our results. This kind of feedback is invaluable in ensuring the accuracy and
reliability of scientific communication. If you have any further questions or need additional
clarification, please don't hesitate to ask.

Figure. 22: Modification on Fig.2e

Comment 6:

The authors conducted ablation experiments on the kNN module and demonstrated that the kNN module does not significantly improve the model. If this module is removed from the model, will it speed up the method or reduce memory usage? Based on the results, there seems to be no necessity to retain this module, let alone use the model with kNN as the default model. Finally, in the caption of Fig. F3, the authors conclude: “And we believe this may be because of the non-biological information within the k-NN graph.” This is undoubtedly quite absurd. If the authors want to analyze the role of the kNN module in detail, they need to supplement more experiments, such as ablation experiments using only kNN.

Answer 6:

Thank you so much for your comments and mentioning the corresponding problems. We sincerely appreciate your careful examination of our figures and tables.

Firstly, we would like to clarify that the observation that the kNN module does not significantly improve the model is an average phenomenon (average over different datasets). As shown in Fig. 3, certain datasets (e.g., darmanis) demonstrate improved results with the inclusion of kNN, while others, such as campbell, indicate a detrimental effect. This variability is indeed an interesting phenomenon. We hypothesize that this may be due to some misinformation in the constructed kNN graph, although we acknowledge that verifying this hypothesis is challenging.

Secondly, incorporating the kNN graph does lead to increased speed and memory usage. . Considering the potential performance benefits, we have decided to keep this module as an optional feature for scGraphformer (results in the experiments verify that in some cases the performance is better with kNN graph).

Thirdly, the kNN module is included in this paper to provide readers with a comprehensive understanding of scGraphformer's capabilities. While our experiments show that the kNN module does not have a significant impact, other researchers with high-quality initial graphs may find this module beneficial. We believe that including this module in the paper would not affect the main contributions of our work.

Thank you once again for your valuable feedback.

**Minor Comments:**

**Comment 1:**

The case of dataset names should be consistent throughout the text (e.g., "campbell" in Fig.
2a and "Campbell" in line 179).

**Answer 1:**

Thank you for your valuable feedback. We have revised the manuscript to ensure consistent
case usage for dataset names throughout the text. We appreciate your attention to detail.

**Comment 2:**

In Table C2, why are the errors for the scmap-cell method and the scmap-cluster method
consistently 0? Additionally, the authors need to standardize the number of decimal places in
the table.

**Answer 2:**

Thank you for your insightful comment and this problem is happened because we set a fixed
random seed while we split our datasets in our previous experiment in scmap-cell and scmap-
cluster. We have noticed this problem in our revision and evaluated all scmap-cell and
scmap-cluster experiments. The updated experiment results have been added to our
supplementary material. Thanks again for letting us notice this problem.

**Comment 3:**

Why are the error bars in Fig. 3 extending outside the plot?

**Answer 3:**

Thank you for your observation regarding the error bars in Fig. 3. The error bars extending
outside the plot are due to the large variance in some of the measurements. This occurs when
we have a high mean value combined with significant variability in the data. For example,
consider a case where we have a mean value of 0.95 with a standard deviation of 0.1. In a
typical error bar representation (mean \pm standard deviation), this would result in an upper
bound of 1.05 (0.95 + 0.1), which exceeds the maximum value of 1.0 on our scale.

**Comment 4**

Abbreviation issues: each abbreviation should appear alongside its full name only the first
time it is used in the text (e.g., line 20 and line 58).

**Answer 4:**

Thank you for your insightful comment regarding the use of abbreviations. We have reviewed
the manuscript and ensured that each abbreviation is accompanied by its full name the first
time it appears in the text, including GNN, scRNA-seq, kNN. We appreciate your attention to
this detail.

**Comment 5::**

The full name of the same entity throughout the text must be consistent (e.g., line 20 and line
58, line 18 and line 37).

**Answer 5:**
Thank you for your careful observation regarding the consistency of entity names. We have
revised the manuscript to ensure that the full name of the same entity is used consistently
throughout the text, specifically on lines 20, 58, 18, and 37. We appreciate your feedback,
which has helped improve the clarity of our work.

**Comment 6::**
Non-proper nouns do not need to be capitalized (line 58).

**Answer 6:**
Thank you for your helpful comment regarding capitalization. We have corrected the text to
ensure that non-proper nouns are not capitalized, particularly on line 58. Your attention to
detail is greatly appreciated and has contributed to the clarity of the manuscript.

**Comment 7:**
The abbreviations of the same entity throughout the text must be consistent (e.g., "GNN;
GNNs" and "k-NN; kNN; KNN").

**Answer 7:**
Thank you for your valuable feedback regarding abbreviation consistency. We have reviewed
the manuscript and ensured that the abbreviations for the same entity are used consistently
throughout the text, specifically standardizing "GNN; GNNs" and "k-NN; kNN; KNN." Your
attention to this detail has significantly improved the clarity of our work.

**Comment 8:**
Each abbreviation must have a full name (e.g., GCN and t-SNE).

**Answer 8:**
Thank you for your important suggestion regarding the inclusion of full names for
abbreviations. We have revised the manuscript to ensure that each abbreviation, including
GCN (Graph Convolutional Network) and t-SNE (t-distributed Stochastic Neighbor
Embedding), is accompanied by its full name upon first use. Your input has helped enhance
the clarity of our work.

**Comment 9:**
There are citation errors for Fig. 5d and Fig. 5e (lines 299 and 310).

**Answer9:**
Sorry for making such errors and we have fixed the problems.

**Comment 10:**
The y-axis label of Fig. 5a should not be "cell type numbers."

**Answer 10:**
Thank you for your observation regarding the y-axis label of Fig. 5a. We have updated the
label to better reflect the intended meaning. Your feedback has been instrumental in
improving the clarity of our figures.

Figure. 23: Modification on cell type number distribution in Zheng 68K dataset

Comment 11:

In Fig. 5b, there should be "scVI" instead of "scvi."

Answer 11:

Thank you for pointing out the labeling issue in Fig. 5b. We have corrected it to display "scVI" instead of "scvi." Your attention to detail is greatly appreciated and helps enhance the quality of our manuscript.

Figure. 24: Modification on UMAP annotation in Fig. 5

Comment 12:

Line 480 lacks a citation for the Klein dataset.

Answer:

Thank you for bringing to our attention the missing citation for the Klein dataset on line 480. We have now added the appropriate citation to ensure proper attribution. Your feedback has been invaluable in improving the accuracy of our manuscript.

Comment 13:

What do the arrows in Figure 5c represent?

Answer 13:

Sorry for confusing you, the arrows point out the cell number of each tissue. It shows that Covid Atlas is constituted by cells from the three tissues.

**Comment 14:**

The authors need to carefully review the entire manuscript to avoid similar errors.

Answer 14:

Thank you for your constructive feedback. We acknowledge the importance of thorough
review and have conducted a comprehensive examination of the entire manuscript to identify
and rectify any similar errors. We appreciate your guidance in helping us enhance the quality
of our work.

**Review of the codes:**

**Comment 1:**

Despite minimal usage instructions, a more detailed tutorial is needed.

Answer 1:

Thank you for your feedback. We've added a more detailed guide on how to use our code.
We also simplified some steps to make it easier for users. You can find the updated code at
the same GitHub address (<https://github.com/xyfan22/scGraphformer>). Thanks again for your
help in improving our project.

**Comment 2:**

The installation process is somewhat complex. The authors have provided many required
packages, but it seems that not all of them are necessary. For example, the rpy2 package may
not need to be installed.

Answer 2:

Thank you for your valuable feedback on our installation process. You're right, and we
appreciate you pointing this out. We've reviewed the package requirements and have
streamlined the list to include only the essential packages needed to run scGraphformer.
We've removed unnecessary dependencies like rpy2 to simplify the installation process. We
hope this makes it easier for users to get started with our tool. If you have any more
suggestions for improvement, please let us know.

**Comment 3:**

There are errors in the demo. It should be "--rand_split" instead of "-rand_split." Also, if set
"-- runs 0" according to the demo, it will result in an error (which seems inconsistent with the
default value of 1 in the code).

Answer 3:

Thank you for spotting these mistakes in our demo. We have fixed these bugs and errors in
our demo. It should be "--rand_split" not "-rand_split". Thanks for helping us make our tool
better. If you see any other problems, please let us know.

**Comment 4:**

Even after overcoming the above difficulties, I still encountered the error "NoneType has no
attribute 'mean'".

Answer 4:

Thank you so much for noticing that, and we found this error is caused by issues happened

in our `logger.print_statistic()` function and we have fixed this problem. Thanks again.